# Responses of runoff to historical and future climate variability over China

Chuanhao Wu[1], Bill X. Hu[1,2*], Guoru Huang[3,4], Peng Wang[1], and Kai Xu[1]

[1] Institute of Groundwater and Earth Sciences, Jinan University, Guangzhou 510632, China.
[2] Department of Earth, Ocean and Atmospheric Sciences, Florida State University, Tallahassee, FL, 32306, USA.
[3] School of Civil Engineering and Transportation, South China University of Technology, Guangzhou 510640, China.
[4] State Key Laboratory of Subtropical Building Science, South China University of Technology, Guangzhou 510640, China.

*Correspondence to*: Bill X. Hu (bill.x.hu@gmail.com)

**Abstract.** China has suffered some of the effects of global warming, and one of the potential implications of climate warming is the alteration of the temporal-spatial patterns of water resources. Based on the long-term (1960–2008) water budget data and climate projections from 28 Global Climate Models (GCMs) of the Coupled Model Intercomparison Project Phase 5 (CMIP5), this study investigated the responses of runoff ($R$) to historical and future climate variability in China at both grid and catchment scales using the Budyko-based elasticity method. Results show that there is a large spatial variation in precipitation ($P$) elasticity (from 1.1 to 3.2) and potential evaporation ($PET$) elasticity (from -2.2 to -0.1) across China. The $P$ elasticity is larger in northeast and western China than in southern China, while the opposite occurs for $PET$ elasticity. The catchment properties elasticity of $R$ appears to have a strong non-linear relationship with the mean annual aridity index and tends to be more significant in more arid regions. For the period 1960–2008, the climate contribution to $R$ ranges from -2.4 % yr$^{-1}$ to 3.6 % yr$^{-1}$ across China, with the negative contribution in northeast China and the positive contribution in western China and some parts of the southwest. The results of climate projections indicate that although there is large uncertainty involved in the 28 GCMs, most project a consistent change in $P$ (or $PET$) in China at the annual scale. For the period 2071–2100, the mean annual $P$ is projected to increase in most parts of China, especially the western regions, while the mean annual $PET$ is projected to increase in all of China, particularly the southern regions. Furthermore, greater increases are projected for higher emission scenarios. Overall, due to climate change, the arid regions and humid regions of China are projected to become wetter and drier in the period 2071–2100, respectively (relative to the baseline 1971–2000).

**Key words:** Runoff; Budyko hypothesis; climate elasticity; climate variability; CMIP5 GCMs; China

## 1 Introduction

Climate change has become increasingly significant (IPCC, 2013), and numerous studies have reported that climate warming is likely leading to the alteration of the hydrological cycle (Oki and Kanae, 2006; Jung et al., 2010). The dynamic properties of the hydrological cycle are governed by the interactions and feedbacks between atmospheric and land surface hydrologic

processes on a catchment scale. The potential consequences of anthropogenic climate change on the hydrological cycle have received significant attention over the last two decades (Wang et al., 2012; IPCC, 2013).

Runoff ($R$), as a commonly adopted indicator of the hydrologic cycle, is critical to human lives and economic activities (Milly et al., 2005). There is a great deal of previous work exploring the impact of climate variations on $R$, with the motivation stemming from the region's vast resources (Christensen et al., 2004; Guo et al., 2009, Piao et al., 2010; Chen et al., 2012; Harding et al., 2012; Wang et al., 2012; Xu et al., 2013b), dangers of flooding (Kay et al., 2006, 2009, 2012; Raff et al., 2009; Liu et al., 2013; Xiao et al., 2013; Wang et al., 2013; Smith et al., 2014; Wu et al., 2014, 2015), and agricultural water uses (Vano et al., 2010). The most common practices in these previous studies are to use the hydrological models driven by the output from Global Climate Models (GCMs) to simulate the hydrological process (e.g., $R$) under future climate change scenarios. However, the key issue faced by such studies is the need to convert coarse resolution GCM outputs to local catchment-scale climatic variables at a higher spatial resolution to serve as the input to a hydrological model (Vano et al., 2015; Wu et al., 2015). The impact assessments are resource intensive and usually subject to uncertainties related to the choice of hydrological model, GCMs, emissions scenarios, and downscaling techniques (Vano et al., 2014, 2015).

With the uncertainty in $R$ due to climate change, simple tools able to provide robust estimates of this impact are essential to support policy and planning decisions. Climate elasticity, as an important indicator, provides a measure of sensitivity of the changes in $R$ due to the changes in climate. Schaake (1990) made the first attempt to introduce the concept of elasticity and related the climate elasticity of $R$ to precipitation ($P$). Since then numerous climate elasticity methods have been developed for evaluating the hydrologic response to climate change all over the world (Schaake, 1990; Dooge et al., 1999; Sankarasubramanian et al., 2001; Milly and Dunne, 2002; Fu et al., 2007; Zheng et al., 2009; Ma et al., 2010; Yang and Yang, 2011; Yang et al., 2014; Vano et al., 2015). Sankarasubramanian et al. (2001) provided a detailed category of climate elasticity methods for modelling climate change impacts. One of the most common methods is to analytically derive the sensitivity of $R$ based on the Budyko hypothesis, due to its clear theory and that it does not rely on a large amount of data (Yang and Yang, 2011). More importantly, the Budyko-based elasticity method can derive the climate elasticity and can also represent the impact of the catchment characteristics through the parameters of the Budyko model. Accordingly, it is widely applied for the assessment of the hydrologic impacts of climate change (Dooge et al., 1999; Zheng et al., 2009; Yang and Yang, 2011; Yang et al., 2014).

China is a vast land, spanning many degrees of latitude with complicated terrain, which results in a large regional variation in climate elasticity. The investigation of the $P$ elasticity of $R$ has been reported in many regions of China, such as the Miyun

Reservoir basin (Ma et al., 2010), Luan River basin (Xu et al., 2013a), the headwater catchments of the Yellow River basin
(Zheng et al., 2009), Poyang Lake basin (Sun et al., 2013), and Hai River and Yellow River basins (Yang and Yang, 2011;
Liu and McVicar, 2012). Recently Yang et al. (2014) investigated the climate elasticity of $R$ for the 210 catchments of China
based on the Budyko-based elasticity approach. The results indicated that the $P$ elasticity exhibits a large regional variation,
with a small range in southern China, the Songhua River basin and the northwest and a large range in the Hai River basin,
the Yellow River basin, and the Liao River basin. Although the aforementioned studies have certainly made advances in
understanding the climate elasticity of $R$ in China, our knowledge about the responses of $R$ to climate change over various
temporal and spatial scales remains rather limited due to the large regional variation in climate types and catchment
characteristics. The question of how climate change will affect $R$ over China in the future is also an important problem to be
addressed. Developing a more accurate and quantitative understanding of the changing water resources over various
temporal and spatial scales under a changing environment is therefore a high priority for China.

Based on the unique long-term (1960–2008) land surface dataset of China and the climate projections from 28 GCMs of the
Coupled Model Intercomparison Project Phase 5 (CMIP5), the objectives of this research are (1) to investigate the changes
of $R$ and climate variables and their relationship at an interannual scale; (2) to estimate quantitatively the climate elasticity
and catchment properties elasticity of $R$ across China at both grid and catchment scales; and (3) to predict climate change
and the changes in $R$ due to future climate change for China from the CMIP5 projections at both grid and catchment scales.

**2 Data and methodology**
**2.1 Data sets**
Monthly data of potential evaporation ($PET$) covering the period 1960–2008 over China are provided by the
Hydroclimatology Group of Princeton University (Sheffield et al., 2006, 2012). The $PET$ is estimated by the Penman
equation (Penman, 1948; Shuttleworth, 1993), using the updated meteorological dataset obtained from Sheffield et al. (2006,
2012). A long-term (1960–2008) daily land surface dataset over China, including $P$, surface runoff ($RS$), and baseflow ($BS$),
with a 0.25 degree spatial resolution were obtained from the Land Surface Processes and Global Change Research Group
(Zhang et al., 2014). In this dataset, $P$ is driven by interpolating gauged daily precipitation from 756 meteorological stations
of the Chinese Meteorological Administration (CMA). $RS$ and $BS$ are derived from the Variable Infiltration Capacity (VIC)
model forced by the gridded daily climate forcings (i.e. $P$, maximum and minimum temperature, and wind speed). VIC
model parameters, including the infiltration shape parameter, the second and third soil layer depths, and the three parameters
in the base flow scheme, were estimated by using an optimization algorithm of the multi-objective complex evolution of the
University of Arizona (Zhang et al., 2014). The simulated monthly $RS$ and $BS$ match well with the observations at the large
river basins in China (Zhang et al., 2014). Compared with the global product of a similar nature, this dataset provides a more
reliable estimate of land surface variables over China (Nijssen et al., 2001; Adamet al., 2006; Rodell et al., 2004; Sheffield et
al., 2006; Sheffield and Wood, 2007; Pan et al., 2012). In this study, the data of $P$, $RS$, and $BS$ are initially regridded onto 0.5°
grids over China using the linear interpolation method. All the daily data ($P$, $RS$, and $BS$) and monthly data ($PET$) are then
aggregated temporally for the annual scale over China. The $R$ was calculated by the sum of $RS$ and $BS$ at each of the 0.5° grid
points.

Climate projections from 28 CMIP5 GCMs (as shown in Table 1) are provided by the Canadian Climate Data and Scenarios
(CCDS, http://www.cccsn.ec.gc.ca/index.php?page=gridded-data). These data, including simulations of surface air
temperature ($T$), $P$, sea ice thickness, sea ice concentration, snow depth, and near-surface wind speed, are statistically
downscaled and regridded onto a common 1°×1° global grid by the CCDS. In this study, monthly $P$ and monthly $T$ over
China, including one historical simulation for the period 1971–2000 and three emission scenarios (RCP2.6, RCP4.5, and
RCP8.5) for the future period 2071–2100 from each of the 28 CMIP5 models and the multi-model ensemble of 28 CMIP5
models, are used for the projections of climate change. The data are initially disaggregated to 0.5° grids over China then
corrected by using a 'delta change' method (Wu et al., 2016), on the basis of the observed data of $P$ and $T$ as provided by the
Climatic Research Unit (CRU) of the University of East Anglia (Harris et al., 2014).

Figure 2 shows the comparison of observed mean annual $T$ and $P$ and the corresponding simulations from 28 CMIP5 models
before and after bias correction for the 14 basins in China. The basin number is consistent with that given in Figure 1. As
shown, the uncorrected model simulations tend to underestimate $T$ and overestimate $P$ for most of the basins, with more
uncertainties for the simulation of $P$ than for $T$. Compared to the uncorrected model results, the bias correction results
represent large improvements and show a good agreement with the observed values for these basins. Therefore, the bias
correction model simulations are acceptable for the investigation of climate change projections in this study.

As the GCM data used only consist of $P$ and $T$, the $PET$ of GCM is estimated by the Thornthwaite method (Thornthwaite,
1948) and then corrected by a multiplicative bias correction method as follows:

$$PET_{cor,GCM,i} = PET_{Th,GCM,i} \times \frac{\overline{PET}_{Pen,obs,i}}{\overline{PET}_{Th,obs,i}} \tag{1}$$

where $PET_{cor,GCM,i}$ and $PET_{Th,GCM,i}$ are bias-corrected annual $PET$ and the $PET$ calculated from the Thornthwaite method,
respectively, for the $i$th grid point of the GCMs. $\overline{PET}_{Pen,obs,i}$ and $\overline{PET}_{Th,obs,i}$ are the 49-year (1960–2008) averages of
$PET$ calculated from the Penman and Thornthwaite methods, respectively, for the $i$th grid point.

Based on the $T$ data from the CRU, the Thornthwaite method is used to calculate $PET$ to test the applicability of Equation (1).

Figure 3 shows a comparison of annual $PET$ calculated from the Penman method and that from the Thornthwaite method

corrected by Equation (1) during the period 1960–2008. It is clear that the corrected $PET$ agrees well with the $PET$ from the

Penman method, with the correlation coefficients of 0.94 and 0.958 at the catchment and grid scales, respectively. This

suggests that Equation (1) can be acceptable for the bias correction of $PET$ in the GCMs.

## 2.2 Sensitivity of runoff to climate and catchment properties

The Budyko framework has been widely used to study basin-scale water and energy balances. Two of the one-parameter

formulations of the Budyko curve proposed by Choudhury (1999) (Equation (2), see also Yang et al., 2008) and Fu (1981)

(Equation (3), see also Zhang et al., 2004) are expressed as:

$$E = P \frac{PET}{(P^n + PET^n)^{1/n}}, \quad n \in (0, \infty) \tag{2}$$

$$E = P + PET - (P^\omega + PET^\omega)^{1/\omega}, \quad \omega \in (1, \infty) \tag{3}$$

where $n$ and $\omega$ are empirical parameters, representing the effects of other factors (e.g. land surface characteristics, the

average slope, vegetation type or land use, and climate seasonality) on the water-energy balance (Yang et al., 2008, 2014;

Roderick and Farquhar, 2011; Li et al., 2013a). Yang et al. (2008) calibrated the parameters $n$ and $\omega$ using long-term water

balance data from 108 catchments from the nonhumid regions of China and found that these two empirical parameters are

linearly correlated.

Based on the Budyko hypothesis and assuming steady state conditions, Roderick et al. (2011) and Yang and Yang (2011)

derived the elasticity method to estimate the contribution to $R$ from the changes in climate (represented by $P$ and $PET$) and

catchment properties as follows:

$$\frac{dR}{R} = \varepsilon_P \cdot \frac{dP}{P} + \varepsilon_{PET} \cdot \frac{dPET}{PET} + \varepsilon_n \cdot \frac{dn}{n} \tag{4}$$

where $\varepsilon_P, \varepsilon_{PET},$ and $\varepsilon_n$ represent the elasticity coefficients of $P$, $PET$, and catchment properties respectively, and are

expressed as:

$$\varepsilon_P = \frac{P}{R}(1 - \frac{\partial E}{\partial P}) \tag{5a}$$

$$\varepsilon_{PET} = -\frac{PET}{R} \frac{\partial E}{\partial PET} \tag{5b}$$

$$\varepsilon_n = -\frac{n}{R}\frac{\partial E}{\partial n} \tag{5c}$$

where $\dfrac{\partial E}{\partial P}$, $\dfrac{\partial E}{\partial PET}$, and $\dfrac{\partial E}{\partial n}$ denote the first order partial derivatives of the Budyko equation with respect to $P$, $PET$, and

the parameter $n$. In this study, both Equations (2) and (3) are used for the estimation of the elasticity of $P$, $PET$, and

catchment properties over China.

**2.3 Trend estimate method**

The Mann-Kendall (M-K) nonparametric test (Mann, 1945; Kendall, 1975) is an effective tool for detecting the statistical

significance of trends in the time series of meteorological and hydrological variables (Yang et al., 2014; Wu and Huang,

2015). In this study, the M-K method is used to detect the significance of monotonic trends in hydroclimatic time series. The

nonparametric trend slope estimator developed by Sen (1968) is used for the magnitude estimation of the trends in a

hydroclimatic time series.

**3 Results**

**3.1 Interannual variability of climatic variables and runoff**

The standard deviations for annual $P$, $PET$, and $R$ are computed for each of the 0.5° grids in China, and the $PET$ deviation

ratio ($\sigma_{PET}/\sigma_P$) and the $R$ deviation ratio ($\sigma_R/\sigma_P$) are calculated. The spatial distributions of $PET$ deviation ratio and $R$

deviation ratio across China are displayed in Figure 4(a) and (b). As shown, the $PET$ deviation ratio is rather small in most

parts of China, especially the southern regions, while a larger value is observed mainly in the Xinjiang region, where there

are greater aridity indices. Generally, atmospheric water is enough to accommodate the limited $PET$ in humid climates,

which would lead to a limited response of $PET$ to $P$ variability. Specifically, the interannual variability of $PET$ is more

sensitive to that of $P$ in arid climates (with water limits) than in humid climates (with energy limits). In contrast to the $PET$

deviation ratio, the $R$ deviation ratio tends to increase from arid climates to humid climates. The reason for this is that, in arid

climates, the catchment water supply is very limited and gives priority to evaporation and soil storage capability, which leads

to little variation in $R$.

Figure 4(c) shows the relationship between the $R$ deviation ratio and mean annual aridity index ($\bar{\phi}$) for all 0.5° grids in

China. As indicated, $\bar{\phi}$ is a major control for the $R$ deviation ratio under not very dry conditions (e.g. $\bar{\phi} < 10$); that is, the $R$

deviation ratio decreases with increased $\bar{\phi}$. However, under very dry conditions (e.g. $\bar{\phi} > 10$) the $R$ deviation ratio becomes

insensitive to $\overline{\phi}$, since in this case, other factors, such as soil storage capacity, can also have a large impact on the variation
of $R$.

**3.2 Sensitivity of runoff to climate and catchment properties**
**3.2.1 Climate elasticity**
The $P$ elasticity and $PET$ elasticity of $R$ based on Equations (2) and (3) are estimated at each of the 0.5º grids in China. As
shown in Figure 5, the spatial patterns of $P$ elasticity and $PET$ elasticity from Equations (2) and (3) are almost the same in all
regions of China. There is a large spatial variation in $P$ elasticity and $PET$ elasticity, i.e. ranging from 1.1 to 3.2 and from
-2.2 to -0.1 across China, respectively. In particular, $P$ elasticity is more significant in the northeast and western areas than in
southern China, which is in contrast to $PET$ elasticity. Figure 6 shows the relationship between $\overline{\phi}$ and climate ($P$ and $PET$)
elasticity. As shown, the $P$ ($PET$) elasticity first increases (decreases) and then decreases (increases) with the increase of $\overline{\phi}$
under not very dry conditions (i.e. $\overline{\phi}$ <10). However, when $\overline{\phi}$ becomes large enough (e.g. $\overline{\phi}$ >10), both $P$ and $PET$
elasticity becomes insensitive to $\overline{\phi}$.

The climate elasticity estimated for each of the 14 large basins is shown in Table 2. The values of $P$ elasticity are in the range
of 1.39–2.28, with a larger (~smaller) elasticity in the Haihe River and Inner Mongolia River (Southwest Drainage). A
similar phenomenon is found for $PET$ elasticity, which suggests that Haihe River (Southwest Drainage) is the most (least)
sensitive to $PET$ among the 14 basins. Overall the values of $P$ elasticity and $PET$ elasticity derived by Equation (2) are very
close to those from Equation (3), but the difference between them tends to be larger for dry basins with increasing aridity
indices.

By using the estimates of climate elasticity derived by Equation (2), the change in $R$ as a function of the percentage change
in $P$ and $PET$ is calculated for the 14 basins (Figure 7). The $R$ is positively related to $P$ and negatively related to $PET$, and the
magnitudes and patterns of the response of $R$ to changes in $P$ and $PET$ vary in different scales. Generally, the $R$ is more
sensitive to climate in the Haihe River and Inner Mongolia River, while relatively weak sensitivity is found in the Southwest
Drainage and Yangtze.

**3.2.2 Catchment properties elasticity**
The spatial distributions of catchment properties elasticity from Equations (2) and (3) are displayed in Figure 5(e) and (f). As
shown, the catchment properties elasticities for these two equations are rather similar across China, and the values of
Equation (3) are generally smaller than those from Equation (2). Regarding the spatial pattern, the catchment properties
elasticity is very weak (approximately equal to 0) in southern China and some regions of northeast China, but it tends to be
more significant in some water-limited regions of northwest China. Figure 6(c) shows the relationship between $\bar{\phi}$ and the
parameter elasticity for all 0.5° grids in China. It suggests that $\bar{\phi}$ is a major control for catchment properties elasticity
across China, i.e. the catchment properties elasticity would become stronger with increasing aridity indices. The catchment
properties elasticities estimated for the 14 large basins are shown in Table 2. The catchment properties elasticity shows a
large spatial variation, ranging from -2.78 to -0.24 for Equation (2) and from -4.3 to -0.33 for Equation (3). Overall, the
changes in $R$ are more sensitive to catchment properties in arid basins with larger aridity indices, which is consistent with the
findings at the grid scale.

**3.3 Climate change during 1960–2008**
The annual trend magnitudes in $P$, $R$, $PET$, and aridity index during the period 1960–2008 are shown in Figure 8 (a), (b), (c),
and (d). As indicated, both $P$ and $R$ show an increasing trend mainly in the northwest and southeast regions and a decreasing
trend mainly in the central region and North China plain. A significant increasing in $PET$ is detected mainly in northeast
China and eastern China, while the decreases mainly occur in most parts of western China. The aridity index tends to show
an increasing trend in most parts of China, indicating an increasing risk of meteorological drought in these regions during the
past several decades. In contrast, the decrease of aridity index is only found in some parts of western China.

**3.4 Changes in runoff due to climate change during 1960–2008**
Using the estimates of climate elasticity from Equation (2), the contributions of $P$, $PET$, and climate (i.e. $P\& PET$) to $R$ in
China for the period 1960–2008 are calculated (as shown in Figure 8(e), (f), and (g)). A positive contribution (up to 3.7 %
yr$^{-1}$) from $P$ to $R$ is mainly recorded in western China, while a negative contribution is found mainly in northeast China and
North China plain. Negative and positive contributions of $PET$ to $R$ mainly occur in northeast China and western China,
respectively. The contributions of climate, i.e. the sum of the contributions from $P$ and $PET$, ranges from -2.4 % yr$^{-1}$ to 3.6 %
yr$^{-1}$ across China. The spatial pattern of climate is rather similar to that of $P$, showing a negative contribution in northeast
China and a positive contribution in western China and some parts of the southeast. Particularly, the largest positive
contribution of climate occurs in the Tibetan plateau. The contributions of $P$, $PET$, and climate (i.e. $P\& PET$) to $R$ in the 14
river basins for the period 1960–2008 are shown in Table 3. A positive contribution of $P$ is detected in Southeast Drainage,
Southwest Drainage, Qiangtang, Qinghai, Xinjiang and Hexi, while an oppoiste contribution is found in other basins. In
contrast, a negative contribution of $PET$ is found in most of the basins (except for Qiangtang and Hexi). In general, there is
an increased $R$ in Southeast Drainage, Southwest Drainage, Qiangtang, Qinghai, Xinjiang and Hexi (from 0.06 to 1 % yr$^{-1}$)
and a decreased $R$ in other basins (from -1.12 to -0.12 % yr$^{-1}$).

**3.5 Future climate change**
Figure 9 shows the uncertainty range of the relative change in mean annual $P$ and $PET$ in the basins for the period 2071–
2100 under the RCP2.6, RCP4.5, and RCP8.5 scenarios as predicted by 28 CMIP5 models (relative to the baseline 1971–
2000). As shown, there is a large difference between different GCMs and emission scenarios, which highlights the
uncertainty inherent in projections of climate change. However, overall $P$ is projected to increase in most of the basins, and
greater increases are projected for higher emission scenarios. Meanwhile, greater increases tend to be projected for more arid
basins, suggesting a decreasing risk of meteorological drought in the future. The average changes (red dotted lines) of mean
annual $P$ for the 14 basins range from 2.4 % to 11.0 % in RCP2.6, from 4.2 % to 16.0 % in RCP4.5, and from 3.1 % to 23.7 %
in RCP8.5. The largest increase in the RCP2.6 and RCP8.5 scenarios is found for the Qinghai River, while the largest
increase in the RCP4.5 scenario is projected for the Hexi River. For $PET$, there is an increase projected in all basins due to
climate warming, with the largest and smallest increases in the RCP8.5 and RCP2.6 scenarios, respectively. However, a large
uncertainty exists among the GCMs, which is similar to that for $P$. Furthermore, the uncertainty range tends to be larger with
higher emission scenarios. The average changes (red dotted lines) of $PET$ for the basins range from 7.0 % to 12.0 % in
RCP2.6, from 13.5 % to 22.2 % in RCP4.5, and from 27.9 % to 49.8 % in RCP8.5. The largest and smallest average
increases are projected for the Pearl River and Qiangtang River, respectively.

Figure 10 displays the multi-model ensemble median relative change in mean annual $P$ and $PET$ in China for the period
2071–2100 (relative to the baseline 1971–2000). The projected changes in $P$ (or $PET$) have a similar spatial pattern for the
three emission scenarios; that is, $P$ is projected to show an increase in western China and the northeast, and $PET$ is projected
to increase significantly in southern China and some parts of the Tibetan plateau, especially for the RCP8.5 scenario. In
addition, note that there are small changes in $P$ and significant increases in $PET$ projected for southern China. This would
result in an increasing risk of meteorological drought in the future.

**3.6 Future changes in runoff due to climate change**
Based on the estimates of elasticity from Equation (2), the percentage changes in the contributions of annual $P$ and $PET$, as
well as climate, to $R$ from the 28 GCMs for the period 2071–2100 are calculated for each of the 14 basins (relative to the
baseline 1971–2000). As shown in Figure 11, the changes in $P$ contribution mainly follow the changes in $P$ (Figure 9). A
positive contribution from $P$ is projected for most of the basins, and larger contributions occur in more arid basins, as well as
in higher emission scenarios. Negative contributions of $PET$ to $R$ are projected for all basins due to the negative coefficients
of *PET* elasticity. Smaller contributions of *PET* are mainly found in the Southwest Drainage. In contrast, larger contributions
are projected mainly in the Huaihe River, Haihe River, and Inner Mongolia River, where the percentage decreases from the
28 models can be up to 25 %, 35 %, and 90 % in the RCP2.6, RCP4.5 and RCP8.5 scenarios, respectively.

Climate change is projected to reduce the *R* in some humid basins, such as the Southeast Drainage and Pearl River, where the
average changes in the three emission scenarios range from -22.83 % to -3.0 % and from -23.6 % to -3.5 %, respectively
(Figure 11 (g), (h) and (i)). For other basins, particularly for arid basins, the *R* is projected to increase due to climate change.
The largest average changes in *R* under the RCP2.6 and RCP4.5 scenarios are found in the Qinghai River (12.85 % and
16.18 %, respectively). For the RCP8.5 scenario, they are found in the Qiangtang River (18.59 %). Note that there is an
obvious decrease in *R* (-17.59 %) projected for the Huaihe River under RCP8.5 scenario, which is mainly caused by the
larger negative contribution of *PET*.

Figure 12 shows the spatial distributions of the relative changes in the contributions of annual *P* and *PET* as well as climate
to *R* in China for 2071–2100. This is based on the CMIP5 multi-model ensemble medians. Compared with the baseline
1971–2000, the increases in *R* due to the changes in *P* are projected in western China and some parts of northern China, and
this phenomenon is particularly significant in the RCP8.5 scenario (up to 60.3 %). In contrast, the changes in *PET* are
projected to reduce the *R* in all of China, with the larger decreases occurring mainly in the North China plain, northeast, and
some parts of western China. Overall, climate change is projected to cause an obvious increase (decrease) of *R* in western
China (southern China) under any emission scenario (Figure 12(g), (h) and (i)). This suggests that the arid regions (humid
regions) in China will become wetter (drier) in the future.

**4 Discussion**
**4.1 The estimation of elasticity**
The Budyko-based elasticity method is applied to quantify sensitivity of runoff to climate and catchment properties across
China. Two Budyko models proposed by Choudhury (1999) and Fu (1981) are used for the comparison of the estimation of
the climate elasticity of *R*. The results suggest that the climate elasticity is insensitive to the Budyko equations. The climate
elasticity of *R* has been estimated in many regions of China. For example, the values of *P* elasticity are estimated as 2.4 for
the Miyun Reservoir basin (Ma et al., 2010), 2.6 for the Luan River basin (Xu et al., 2013a), 2.1 for the headwater
catchments of the Yellow River basin (Zheng et al., 2009), 1.4–1.7 for the Poyang Lake basin (Sun et al., 2013), 1.7–3.1 for
the Hai River basin (Xu et al., 2014), 1.1–2.0 for southern China, the Songhua River basin, and the northwest, 2.1–4.8 for the
Hai River basin, the Yellow River basin, and the Liao River basin (Yang et al., 2014), and 1.6–3.8 for the 63 catchments of

China (Yang and Yang, 2011). In addition, the *PET* elasticity is estimated as -1.04 for the headwater catchments of the Yellow River basin (Zheng et al., 2009) and from -1 to -0.2 for the Poyang Lake basin (Sun et al., 2013). Those results are close to our results for *P* elasticity ranging from 1.1 to 3.2, and for *PET* elasticity ranging from -2.2 to -0.1 in China. It is worth noting that the values of *P* elasticity tend to be larger in the northeast and some parts of western China that are located in arid climates. This is in good agreement with the findings by Sankarasubramanian et al. (2001), which indicated that a larger *P* elasticity occurs in more arid regions. However, some parts of Xinjiang, which is more arid than southern China, have smaller *P* elasticity. Meanwhile, some parts of southern China, which is more humid than other regions in China, have larger *P* elasticity (Figure 5). In addition, the Haihe River basin, located in less arid climates than that of the northwest, shows the largest *P* elasticity in China (Table 2). A similar phenomenon is also introduced in Yang et al. (2014). One of the major reasons for this difference may be attributed to the impacts of human activities that alter the patterns of *R* in these regions. In addition, uncertainties in water budget data, such as the errors in the simulation of *R* and in the estimation of *PET*, may also contribute to this difference.

The comparisons for the estimates of $\varepsilon_n$ and $\varepsilon_\omega$ suggest that although the values of $\varepsilon_n$ and $\varepsilon_\omega$ are mainly dependent on the parameters of Budyko models, the spatial pattern of $\varepsilon_n$ is consistent with that of $\varepsilon_\omega$ at the 0.5° grid points over China (Figure 5(e) and (f)). Yang et al. (2008) indicated that the parameters *n* and $\omega$ from Equations (2) and (3) are linearly correlated. We also conducted a regression analysis of $\varepsilon_n$ and $\varepsilon_\omega$ for all 0.5° grid points over China and found a strong linear correlation between $\varepsilon_n$ and $\varepsilon_\omega$ ($\varepsilon_\omega = 1.7061\varepsilon_n + 0.0986$, $r^2 = 0.96$). In addition, our results show that *R* is more sensitive to catchment properties ($\varepsilon_n$ and $\varepsilon_\omega$) in the more arid regions (Figure 5(e) and (f)). The possible internal connection is that the arid regions with less vegetation coverage and stronger evaporation do not effectively hold the rainfall water that will be evaporated, leading to the smaller proportion of rainfall for *R*.

**4.2 Sensitivity analysis for *PET* calculation methods**

We compare four *PET* calculation methods, including the Penman method, the Thornthwaite method, the FAO-56 Penman–Monteith method (Allen et al., 1998), and the Thornthwaite method corrected by Equation (1), to test the robustness of the *PET* elasticity result subject to *PET* uncertainties. In terms of mean annual *PET* as shown in Figure 13 (a), the Thornthwaite method gives relatively low *PET* among the four methods, especially in arid basins (e.g., Qiangtang, Qinghai, Xinjiang and Hexi). This is in agreement with previous studies, which indicated that the Thornthwaite method tends to underestimate *PET* in the arid areas (Hashemi and Habibian, 1979; Malek 1987; Garcia et al., 2004). In contrast, the mean annual *PET* by the other three methods are quite consistent, especially for the Penman method and the Thornthwaite method corrected by Equation (1). A similar result was also reported by Zeng and Cai (2016), which indicated that estimations of water balance at

both annual and month scales are generally robust under various *PET* calculation methods (not including the Thornthwaite
method). The *PET* elasticity calculations from the four different *PET* data for the 14 river basins are shown in Figure 13(b).
The Thornthwaite method yields stronger *PET* elasticity than other three methods in most of the basins mainly due to the
underestimation of *PET*. However, the other three methods give very similar results in all 14 basins. In summary, the
estimation of *PET* elasticity is robust to the *PET* calculations from the Penman method, the FAO-56 Penman–Monteith
method, and the Thornthwaite method corrected by Equation (1), but is not acceptable for the Thornthwaite method.

In general, the Thornthwaite method corrected by Equation (1) significantly improves the accuracy of *PET* (Figure 3 and
Figure 13(a)). However, it should be emphasized that the Thornthwaite method is an empirical equation that neglects the
effects of atmospheric conditions, such as wind speed, humidity and radiation (McVicar et al., 2012). In addition, the
Equation (1) used for the bias correction of *PET* belongs to a 'delta method' (Graham et al., 2007; Sperna Weiland et al.,
2010), which only considers the average change but ignores the differences in the standard deviation and the coefficient of
variation between the projection and baseline periods (Watanabe et al., 2012). Therefore, a more physically-based *PET*
calculation method (such as the Penman method) needs to be considered to fully understand the *PET* calculation
uncertainties in the projections of climate change.

**4.3 The projections of climate change and runoff**
The hydrological impacts of climate change have been investigated in many regions of China, such as the Hanjiang basin
(Chen et al., 2007; Guo et al., 2009), the catchment of the Loess Plateau (Wang et al., 2013), the Qingjiang River basin
(Chen et al., 2012), the Qiantang River basin (Xu et al., 2013b), the Songhuajiang River basin (Su et al., 2015), the
southeastern Tibetan Plateau (Li et al., 2013b), the Pearl River basin (Yan et al., 2015), the Xin River basin (Zhang et al.,
2016), the sub-catchments of the Yangtze and Yellow River basins (Xu et al., 2011), the Huang-Huai-Hai region (Lu et al.,
2012), and ten major river basins in China (Wang et al., 2012). There is a large uncertainty involved in these impact studies,
which results in a large difference in climate projections. For example, Wang et al. (2012) indicated that the prevailing
pattern of "north dry and south wet" in China will likely be exacerbated under future climate warming. However, the results
of most GCMs in this study suggest that the arid regions and humid regions of China are projected to become wetter and
drier in the future, respectively. The main difference between the two studies is the use of different climate models, emission
scenarios, and time periods. This also demonstrates that the results of climate projections should be taken with caution, since
the regional climate simulations (especially of precipitation) from the GCMs are still not robust at the present stage.

This study focuses on the hydrological change due to climate change (i.e., changes in *P* and *PET*), while the effects of the
variability of catchment properties (e.g., land cover change, groundwater and river water extraction, urbanization, irrigation,
etc.) on the hydrology are overlooked here. Most of the available GCMs lack of key regional feedback processes involving
land use, such as forest plantations, irrigation, and urbanization feedbacks that are critically important throughout China
(Piao et al., 2010). The projected changes in catchment properties therefore need to be involved in the GCMs to account for
their hydrological impacts. In addition, recent studies indicated that plant responses to increasing $CO_2$ tend to keep more
water on land, hence resulting in a greater increase in $R$ (Milly and Dunne, 2016; Swann et al, 2016). That is to say, the
hydrological models (e.g., VIC model), without the schemes of the plant stomatal responses to $CO_2$, would lead to an
underestimation of $R$ under high $CO_2$. Therefore, the implications of plants needing less water under high $CO_2$ should be
included in the assessment of hydrological impacts of climate change.

**4.4 Uncertainties**
Generally, a multitude of sources of uncertainty are involved in the impact assessment of climate change. In this study,
uncertainty mainly comes from the GCMs, emission scenarios, the elasticity method, and the estimation error of the water
budget data. To highlight the uncertainty from the GCMs, the 28 GCMs as produced by different research institutes around
the world, are used for the comparison of climate change projections. There exists a large difference in the projections of $P$
and $PET$ among the 28 GCMs. Particularly, the uncertainty range of $P$ tends to be larger for more arid regions, while the
uncertainty range of $PET$ tends to be larger for more humid regions (Figure 9). This highlights the impact of potential
misleading conclusions if only one climate model were to be used for the impact assessments. The large uncertainty driven
by the GCMs in relation to the hydrological impacts of climate change has been reported in many previous studies (Kay et
al., 2009; Prudhomme and Davies, 2009; Chen et al., 2011; Teng et al., 2012; Liu et al., 2013; Wu et al., 2014, 2015). It is
worth noting that although the projected ranges of $P$ and $PET$ show large variability in various GCMs, most project a
consistent change (i.e. increase) in $P$ and $PET$ for the future period (Figure 9). In contrast, the uncertainty from the emission
scenarios is smaller than that from the GCMs, since the projected changes in $P$ (or $PET$) show a similar pattern under all
emission scenarios (Figure 9). The main difference is that the projected changes tend to be more significant in higher
emission scenarios.

The elasticity equation (i.e. Equation (4)) used in this study is driven from the linear approximation of the Budyko equation
(Equations (2) and (3)) by neglecting the higher order. Such approximation would possibly lead to an uncertainty in the
estimation of climate elasticity. Yang et al. (2014) indicated that the error in estimation of elasticity tends to increase with
increasing changes in $P$ and $PET$, as well as the increased parameter of the Budyko equation. Future research is needed to
quantify the effects of the errors on the estimation of elasticity under various climate conditions.

In addition to uncertainty in *PET* calculation (as discussed in section 4.2), there are also uncertainties associated with the
estimates of other water budget components, such as *R*. As shown in Figure 14, the sensitivity of climate (i.e., *P* and *PET*)
elasticity to *R* varies considerably between basins and tends to be larger in more humid basins. Moreover, *PET* elasticity is
more sensitive to changes in *R* compared with *P* elasticity for all 14 basins. As indicated by Zhang et al. (2014), although the
*R* is realistically estimated for most of the basins (especially for humid basins) in China with a small relative error, there is
still a large relative error for few arid basins in western China due to the lack of meteorological observations. Therefore, the
large errors in simulated *R* of the VIC model may result in large uncertainties in elasticity calculation, particularly in western
China. Also note that some other natural water sources, such as snow and glaciers, which may contribute to *R*, are
overlooked in this study. Lute and Abatzoglou (2014) highlighted the importance of extreme snowfall events in shaping the
interannual variability of the water balance. The melting of snow and glaciers is generally significant at a seasonal time scale
in some high altitude regions of China. Neglecting the effects of snow and glaciers would lead to a bias in the modelling of *R*
for these regions.

**5 Conclusion**
In this study, the Budyko-based elasticity method was used to investigate the responses of runoff to historical and future
climate variability over China at both grid and catchment scales. The climate and catchment properties elasticities of runoff
were estimated based on the long-term (1960–2008) land surface data from Zhang et al. (2014). Twenty-eight GCMs with
three emission scenarios from the CMIP5 were collected for the projections of climate change and its contribution to runoff
in China during the period 2071–2100. The uncertainties associated with the estimates of *PET*, *R*, climate elasticity, as well
as climate projections, are discussed in detail. The main findings are summarised as follows:

(1) The interannual variability of *PET* is more sensitive to that of *P* in more arid regions, while the opposite occurs in the
response of interannual variability of *R* to that of *P*. A large spatial variation exists in *P* elasticity (from 1.1 to 3.2) and *PET*
elasticity (from -2.2 to -0.1) across China. The *P* elasticity is larger in northeast and western China than in southern China,
which is opposite to that of *PET* elasticity. Among the 14 river basins, the Haihe River and Southwest Drainage have the
largest and smallest climate elasticities, respectively. The catchment properties elasticity of *R* is sensitive to mean annual
aridity indices and tends to be stronger in more arid regions with increasing aridity indices.

(2) For the period 1960–2008, the positive (negative) contributions from *P* to *R* are mainly found in western China (northeast
China and North China plain), and the positive (negative) contributions of *PET* mainly occur in western China (northeast

China). Overall, the climate contribution to $R$ ranges from -2.4 % yr$^{-1}$ to 3.6 % yr$^{-1}$ across China during the period 1960–2008, with a negative contribution in northeast China and a positive contribution in western China and some parts of the southwest. The largest positive and negative contributions of climate occur in the Qiangtang and Haihe River basins, respectively.

(3) There is a large uncertainty in climate projections among the 28 GCMs. Moreover, the uncertainty range of the $P$ ($PET$) projection tends to be larger for more arid (humid) regions. However, most of the GCMs project a consistent change in annual $P$ or annual $PET$. For the period 2071–2100, the $P$ is projected to increase in most parts of China, especially the western regions, and the $PET$ is projected to increase in all of China, particularly the southern regions. Furthermore, greater increases are projected for higher emission scenarios. Due to future climate warming, the arid regions and humid regions of China are projected to become wetter and drier in the period 2071–2100, respectively (relative to the baseline 1971–2000).

The results of this study (especially of the climate change projections) should be taken with caution, since uncertainties in the results exist because of several issues, including the different simulations of GCMs, the estimation error of climate elasticity, and the estimation error in the water budget components. A thorough investigation of the uncertainty involved in the hydrologic effects of climate change in China should be considered in future research.

**Acknowledgements**

This research was supported by the Fundamental Research Funds for the Central Universities (Grant No. 21617301) and partly supported by funding from the National Natural Science Foundation of China (Grant No. 41530316) and the National Key Research and Development Program of China (Grant No. 2016YFC0402805).

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

**Table 1.** CMIP5 GCMs used in this study. The GCM data were statistically downscaled and regridded onto a common 1°×1° global grid by the Canadian Climate Data and Scenarios (CCDS).

| No. | Model | Institution (Country) | Resolution |
|-----|-------|----------------------|------------|
| 1 | BCC-CSM1-1 | Beijing Climate Center, China Meteorological Administration, China | 1°×1° |
| 2 | BCC-CSM1-1-m | | |
| 3 | BNU-ESM | College of Global Change and Earth System Science, Beijing Normal University, China | 1°×1° |
| 4 | CCSM4 | National Center for Atmospheric Research, USA | 1°×1° |
| 5 | CESM1-CAM5 | Community Earth System Model Contributors, USA | 1°×1° |
| 6 | CNRM-CM5 | Centre National de Recherches Météorologiques / Centre Européen de Recherche et Formation Avancée en Calcul Scientifique, France | 1°×1° |
| 7 | CSIRO-Mk3-6-0 | Commonwealth Scientific and Industrial Research Organization in collaboration with Queensland Climate Change Centre of Excellence, Australia | 1°×1° |
| 8 | CanESM2 | Canadian Centre for Climate Modelling and Analysis, Canada | 1°×1° |
| 9 | EC-EARTH | EC-EARTH consortium | 1°×1° |
| 10 | FGOALS-g2 | LASG, Institute of Atmospheric Physics, Chinese Academy of Sciences and CESS, Tsinghua University, China | 1°×1° |
| 11 | FIO-ESM | The First Institute of Oceanography, SOA, China | 1°×1° |
| 12 | GFDL-CM3 | | |
| 13 | GFDL-ESM2G | NOAA Geophysical Fluid Dynamics Laboratory, USA | 1°×1° |
| 14 | GFDL-ESM2M | | |
| 15 | GISS-E2-H | NASA Goddard Institute for Space Studies, USA | 1°×1° |
| 16 | GISS-E2-R | | |
| 17 | HadGEM2-AO | National Institute of Meteorological Research/Korea Meteorological Administration, South Korea | 1°×1° |
| 18 | HadGEM2-ES | Met Office Hadley Centre (additional HadGEM2-ES realizations contributed by Instituto Nacional de Pesquisas Espaciais), UK | 1°×1° |
| 19 | IPSL-CM5A-LR | Institut Pierre-Simon Laplace, France | 1°×1° |
| 20 | IPSL-CM5A-MR | | |
| 21 | MIROC-ESM | Japan Agency for Marine-Earth Science and Technology, Atmosphere and Ocean | 1°×1° |
| 22 | MIROC-ESM-CHEM | Research Institute (The University of Tokyo), and National Institute for Environmental Studies, Japan | |
| 23 | MIROC5 | Atmosphere and Ocean Research Institute (The University of Tokyo), National Institute for Environmental Studies, and Japan Agency for Marine-Earth Science and Technology, Japan | 1°×1° |
| 24 | MPI-ESM-LR | Max-Planck-Institut für Meteorologie (Max Planck Institute for Meteorology), Germany | 1°×1° |
| 25 | MPI-ESM-MR | | |
| 26 | MRI-CGCM3 | Meteorological Research Institute, Japan | 1°×1° |
| 27 | NorESM1-M | Norwegian Climate Centre, Norway | 1°×1° |
| 28 | NorESM1-ME | | |

**Table 2**. The estimations of $P$ elasticity, $PET$ elasticity, and catchment properties elasticity of $R$ in the
14 river basins of China based on Equations (2) and (3). The basin number is consistent with that given
in Figure 1. The numbers in the parentheses indicate the 1960–2008 mean aridity index.

| Basin No. | $\varepsilon_P$ | | $\varepsilon_{PET}$ | | $\varepsilon_n$ or $\varepsilon_\omega$ | |
|---|---|---|---|---|---|---|
| | Eq.(2) | Eq.(3) | Eq.(2) | Eq.(3) | Eq.(2) | Eq.(3) |
| 1 (0.52) | 1.64 | 1.65 | -0.64 | -0.65 | -0.24 | -0.33 |
| 2 (0.64) | 1.63 | 1.64 | -0.62 | -0.63 | -0.41 | -0.61 |
| 3 (0.81) | 1.55 | 1.56 | -0.55 | -0.55 | -0.57 | -0.93 |
| 4 (1.19) | 1.40 | 1.39 | -0.40 | -0.39 | -0.73 | -1.44 |
| 5 (1.19) | 2.09 | 2.08 | -1.08 | -1.07 | -1.03 | -1.47 |
| 6 (1.43) | 2.06 | 2.04 | -1.05 | -1.02 | -1.25 | -1.83 |
| 7 (1.71) | 1.92 | 1.88 | -0.91 | -0.87 | -1.35 | -2.10 |
| 8 (2.14) | 2.28 | 2.21 | -1.29 | -1.22 | -1.89 | -2.70 |
| 9 (2.38) | 1.78 | 1.72 | -0.79 | -0.73 | -1.53 | -2.54 |
| 10 (4.41) | 2.23 | 2.11 | -1.22 | -1.10 | -2.78 | -4.16 |
| 11 (4.70) | 1.81 | 1.72 | -0.82 | -0.72 | -2.17 | -3.67 |
| 12 (6.68) | 1.72 | 1.62 | -0.73 | -0.63 | -2.28 | -4.08 |
| 13 (8.09) | 1.66 | 1.56 | -0.65 | -0.55 | -2.26 | -4.27 |
| 14 (8.63) | 1.63 | 1.53 | -0.64 | -0.54 | -2.26 | -4.30 |


**Table 3**. The contributions of $P$, $PET$, and climate (i.e. $P\& PET$) to $R$ in the 14 basins of China for the
period 1960–2008. The basin number is consistent with that given in Figure 1. The numbers in the
parentheses indicate the 1960–2008 mean aridity index.

| Basin No. | $P$ (%/a) | $PET$ (%/a) | $P\&PET$ (%/a) |
|---|---|---|---|
| 1 (0.52) | 0.19 | -0.13 | 0.06 |
| 2 (0.64) | -0.03 | -0.09 | -0.12 |
| 3 (0.81) | -0.07 | -0.07 | -0.14 |
| 4 (1.19) | 0.14 | -0.01 | 0.13 |
| 5 (1.19) | -0.18 | -0.27 | -0.45 |
| 6 (1.43) | -0.35 | -0.31 | -0.66 |
| 7 (1.71) | -0.57 | -0.34 | -0.91 |
| 8 (2.14) | -0.74 | -0.38 | -1.12 |
| 9 (2.38) | -0.38 | -0.04 | -0.42 |
| 10 (4.41) | -0.40 | -0.26 | -0.66 |
| 11 (4.70) | 0.99 | 0.01 | 1.00 |
| 12 (6.68) | 0.43 | -0.01 | 0.42 |
| 13 (8.09) | 0.84 | -0.02 | 0.82 |
| 14 (8.63) | 0.11 | 0.08 | 0.19 |


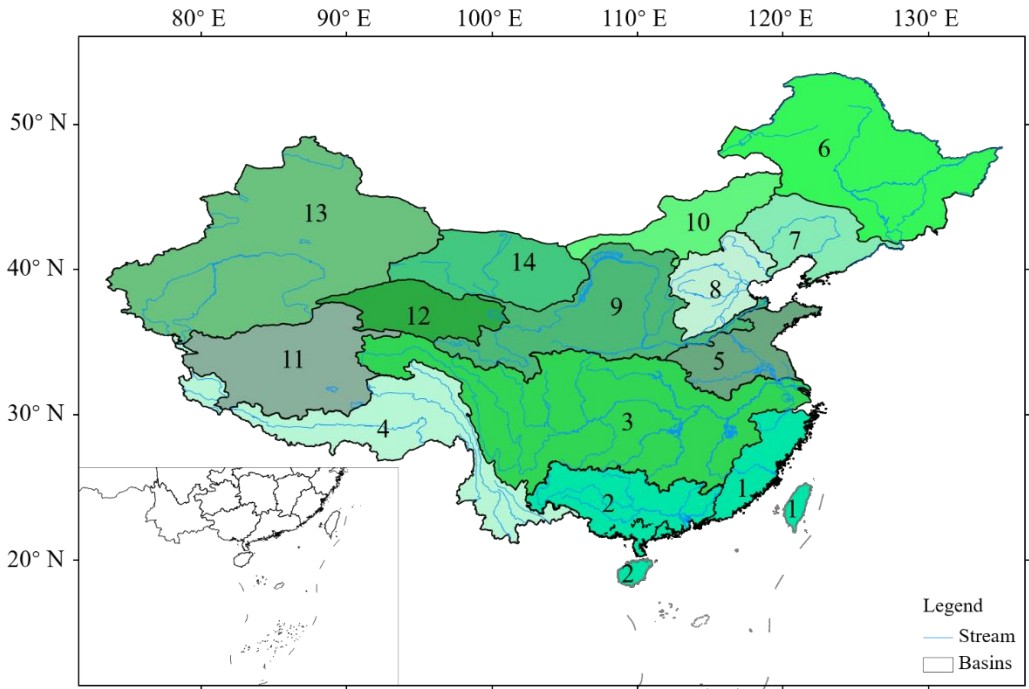

Figure 1. Location of the main river basins in China. The numbers denote the river basins with increasing aridity index: 1, Southeast Drainage (0.52); 2, Pearl River (0.64); 3, Yangtze River (0.81); 4, Southwest Drainage (1.19); 5, Huaihe River (1.19); 6, Heilongjiang River (1.43); 7, Liaohe River (1.71); 8, Haihe River (2.14); 9, Yellow River (2.38); 10, Inner Mongolia River (4.41); 11, Qiangtang River (4.70); 12, Qinghai River (6.68); 13, Xinjiang River (8.09), 14, Hexi River (8.63). The numbers in the parentheses indicate the 1960–2008 mean aridity index.

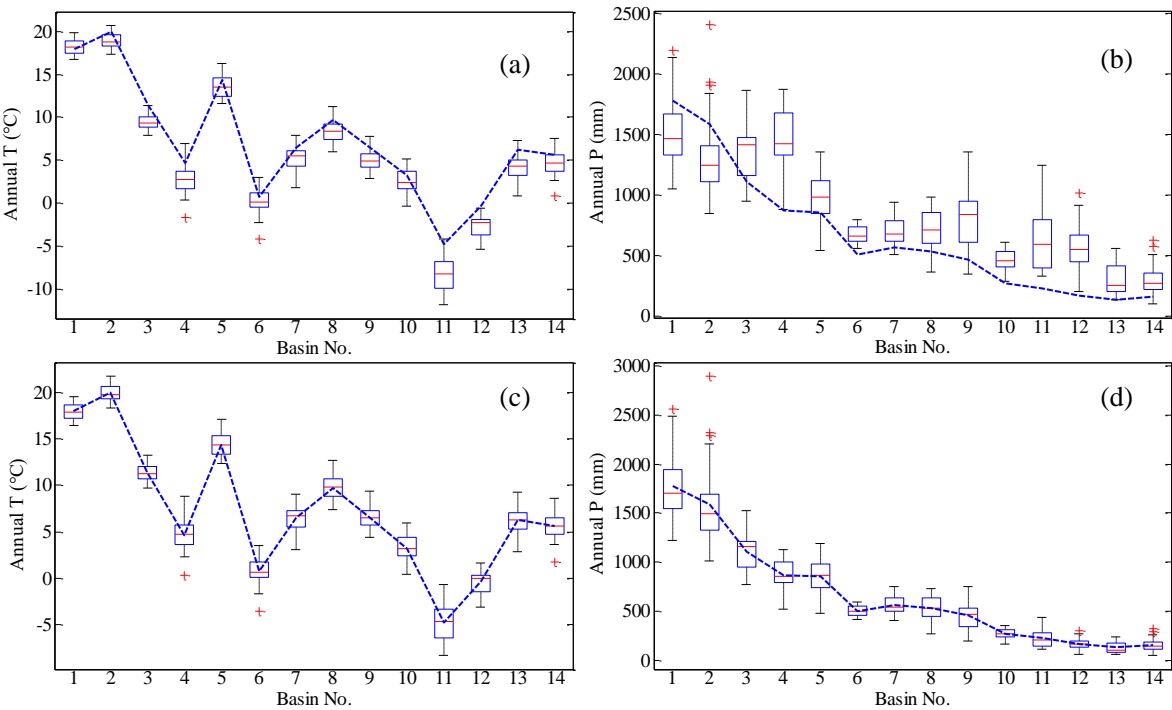

669

**Figure 2**. Box plots of the simulation results of (a) mean annual $T$ and (b) mean annual $P$ and the bias

correction results of (c) mean annual $T$ and (d) mean annual $P$ from 28 GCMs for the period 1971–2000

in the 14 river basins. The boxes denote the interquartile model spread (range between the 25th and 75th

quantiles), with the horizontal line indicating the ensemble median and the whiskers showing the

extreme range of the 28 CMIP5 model simulations. The blue dotted lines denote the observed results of

mean annual $T$ and mean annual $P$ for the period 1971–2000. The basin number is consistent with that

given in Figure 1.


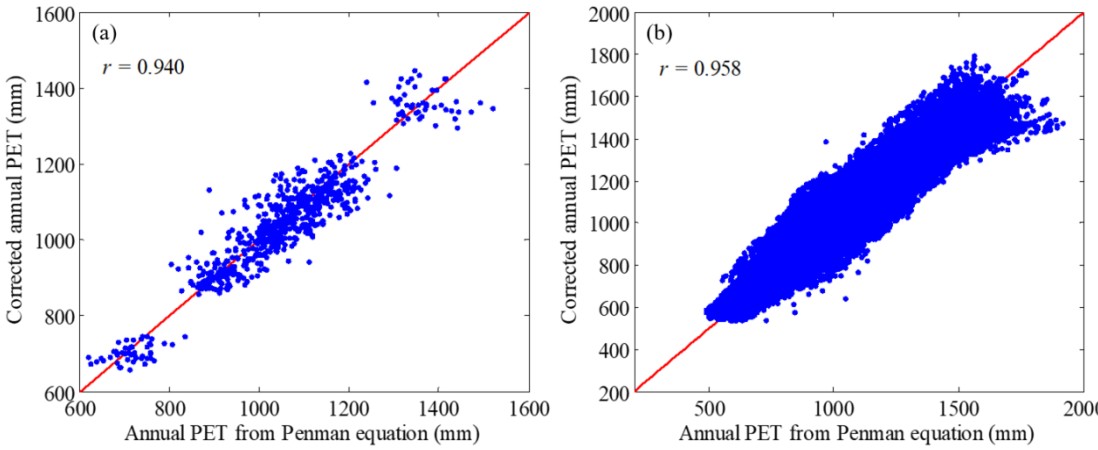


**Figure 3.** Comparison of annual *PET* calculated from the Penman method and the Thornthwaite method
corrected by Equation (1) during the period 1960–2008 for (a) the 14 river basins and (b) all 0.5º grid
points over China.

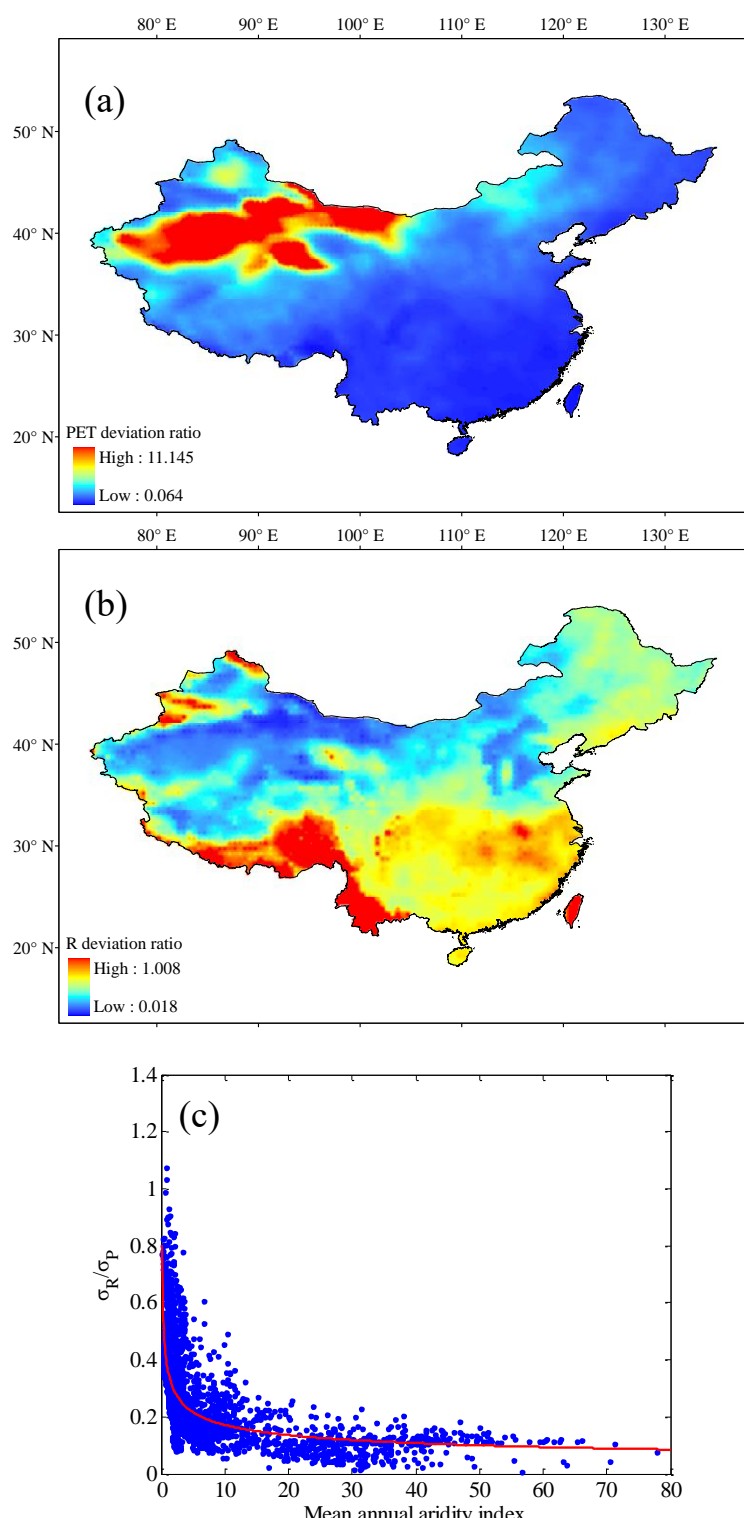


**Figure 4.** Spatial distributions of (a) *PET* deviation ratio and (b) *R* deviation ratio and (c) the

relationship between *R* deviation ratio and mean annual aridity index ($\bar{\phi}$) for all 0.5º grid points in

China.

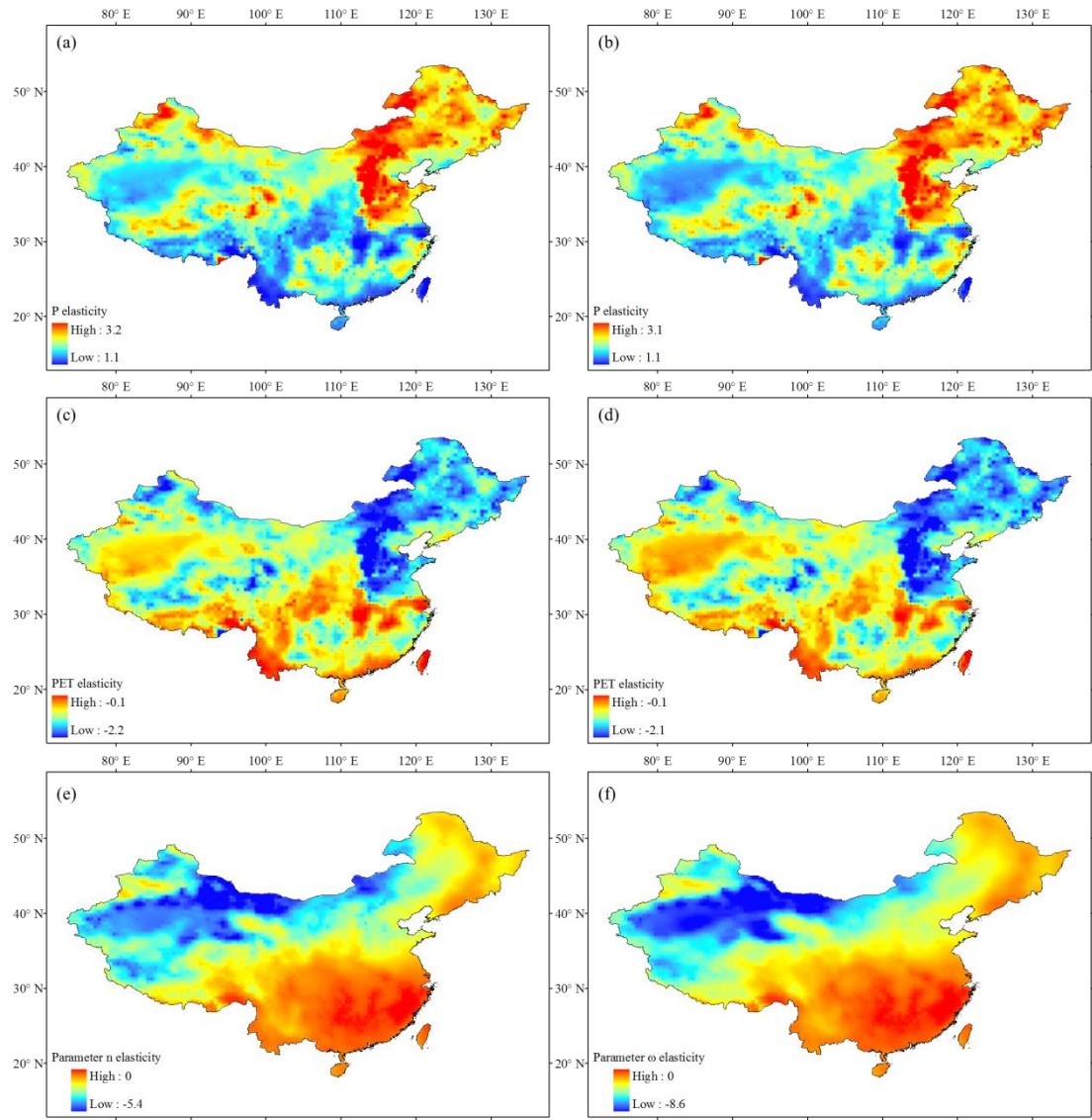

**Figure 5**. Spatial distributions of the *P* elasticity of *R* across China from (a) Equation (2) and (b)

Equation (3). Spatial distributions of the *PET* elasticity of *R* across China from (c) Equation (2) and (d)

Equation (3). Spatial distributions of the parameter elasticity of *R* across China from (e) Equation (2)

and (f) Equation (3).

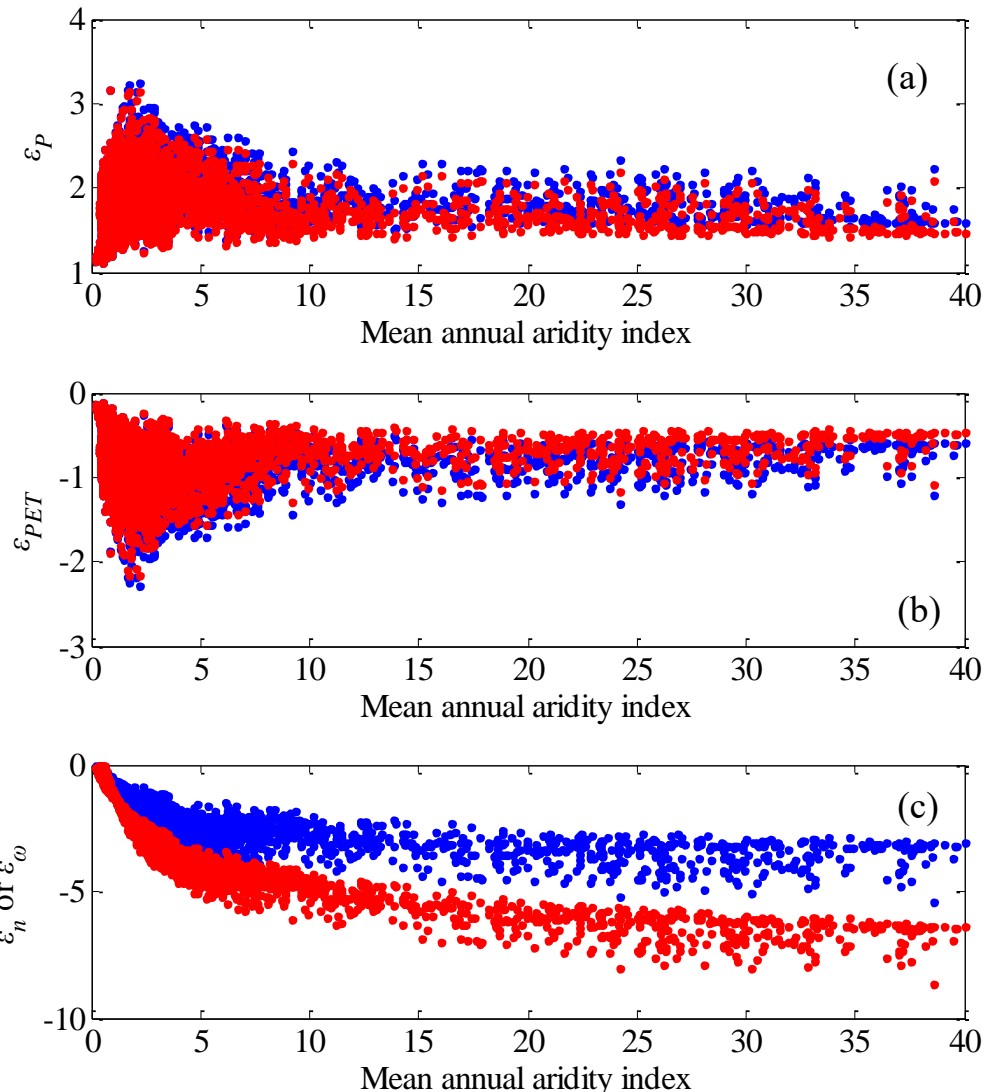


**Figure 6.** The relationship between mean annual aridity index and (a) *P* elasticity, (b) *PET* elasticity,

and (c) parameter elasticity. The blue points represent the case of Equation (2), and the red points

represent the case of Equation (3).


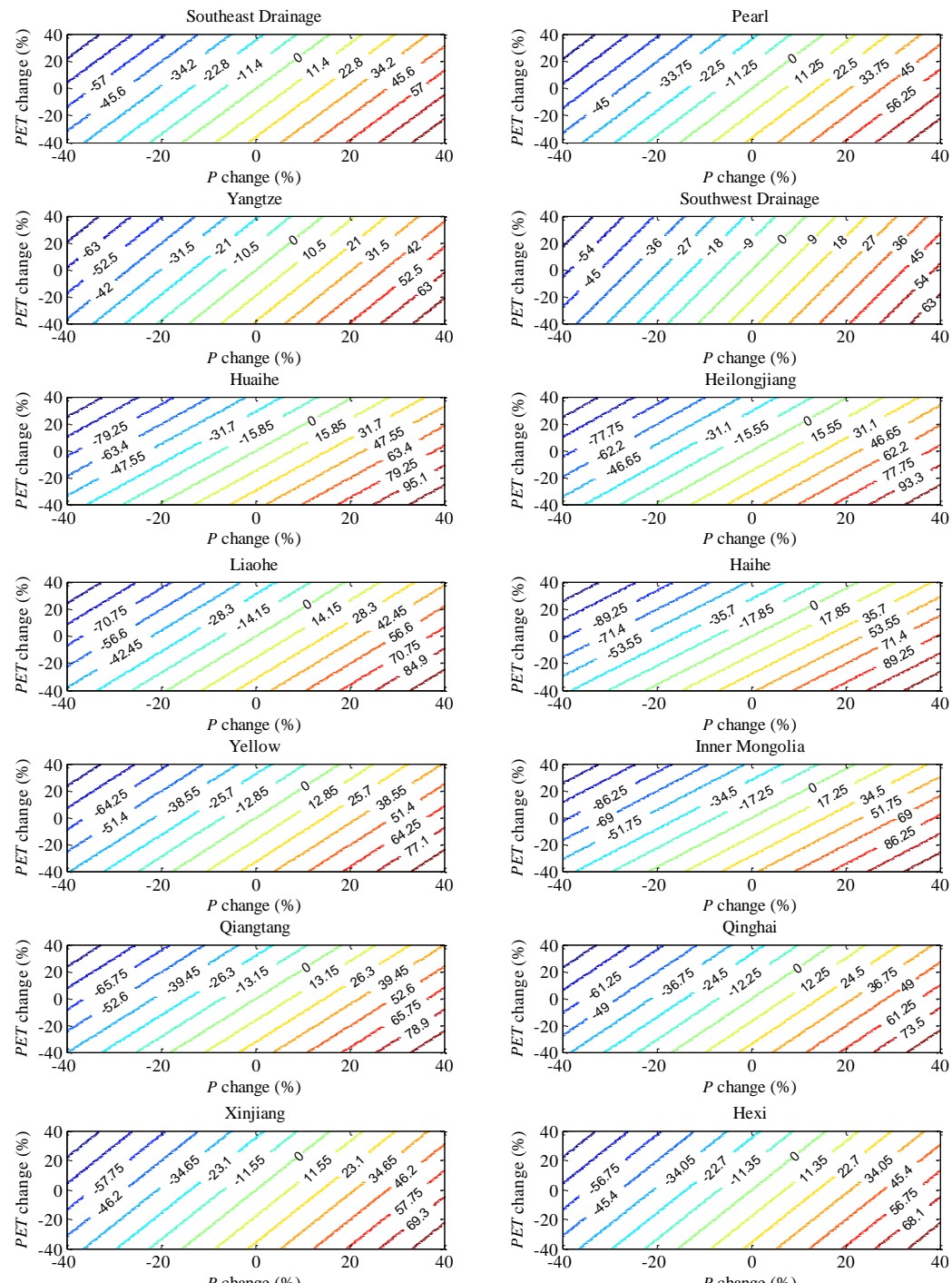


**Figure 7**. Contour plot of percentage $R$ change due to the changes in $P$ and $PET$ for the 14 river basins.

The $P$ elasticity and $PET$ elasticity of $R$ are estimated based on Equation (2).



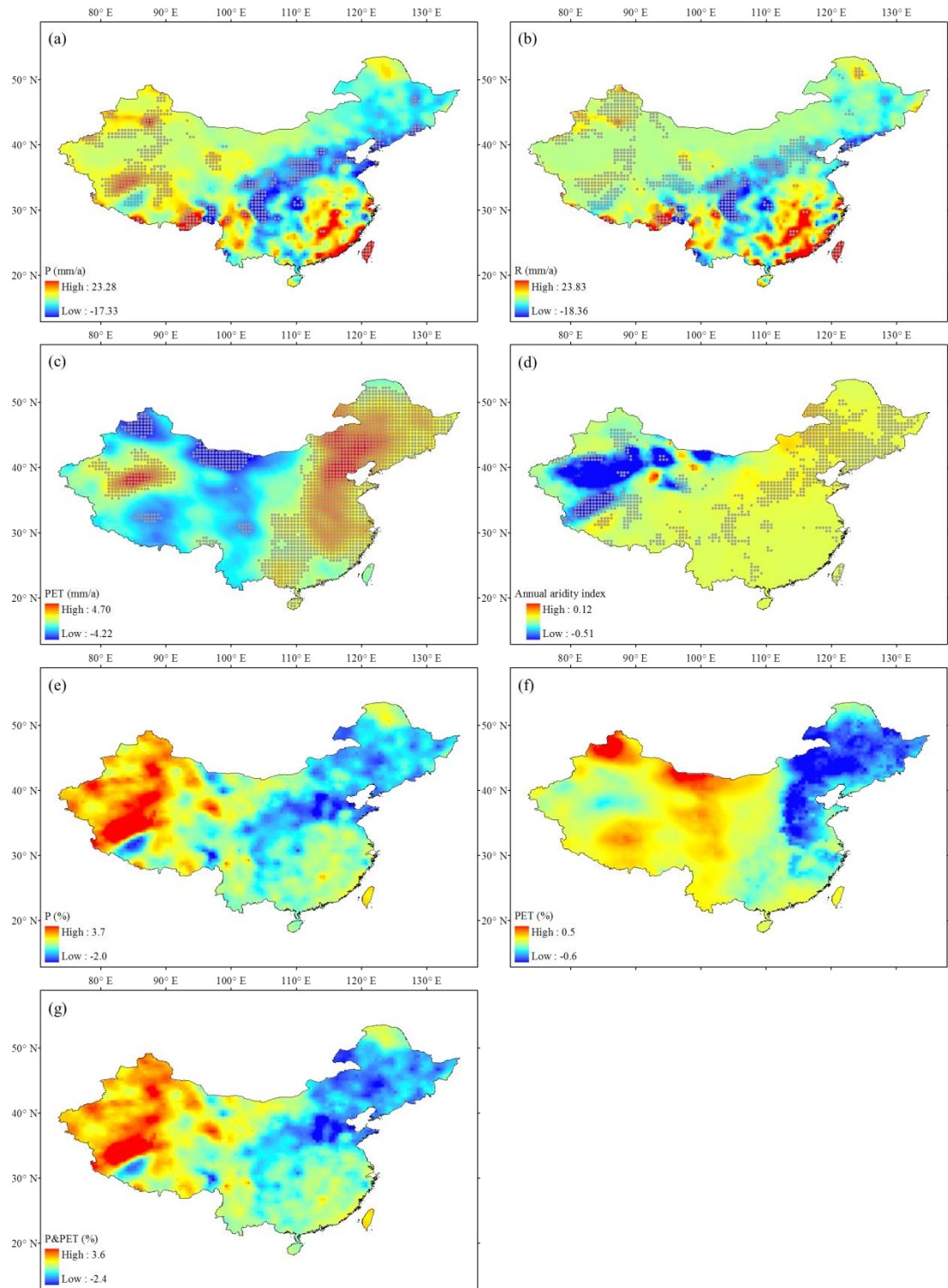


**Figure 8.** Trend magnitudes in annual time series of (a) *P*, (b) *R*, (c) *PET*, and (d) aridity index for the

period 1960–2008 and spatial distributions of the contributions (unit: % yr⁻¹) of (e) *P*, (f) *PET*, and (g)

climate (i.e. *P& PET*) to *R* in China for the period 1960–2008. The trend magnitudes are estimated by

the Sen's method. Grey dots are shown as statistically significant positive/negative trends (*p* < 0.05).

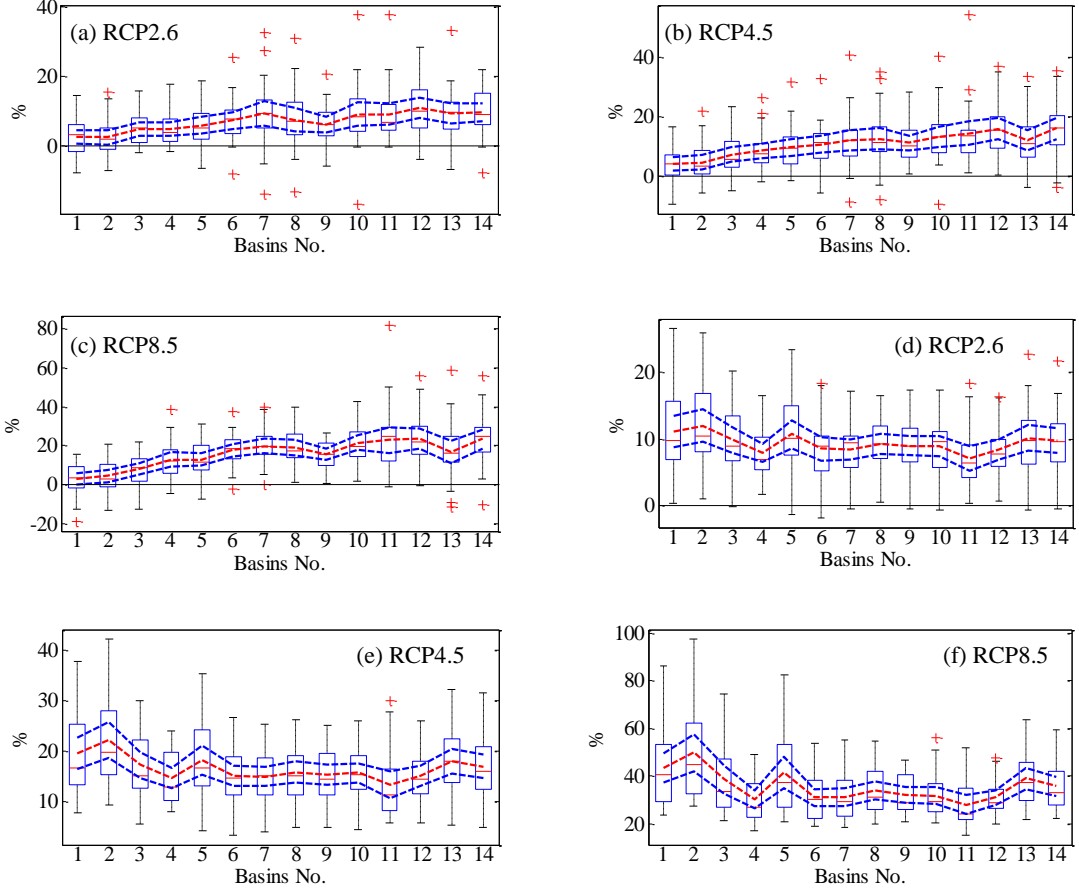

**Figure 9.** Box plots of relative change (%) in mean annual *P* under (a) RCP2.6, (b) RCP4.5, and (c) RCP8.5 scenarios and in mean annual *PET* under (d) RCP2.6, (e) RCP4.5, and (f) RCP8.5 scenarios calculated from 28 CMIP5 models in 14 basins for the period 2071–2100 (relative to the baseline 1971–2000). The boxes denote the interquartile model spread (range between the 25th and 75th quantiles), with the horizontal line indicating the ensemble median and the whiskers showing the extreme range of the 28 CMIP5 model simulations. Red dotted lines denote the average values of the multi-model ensemble. Blue dotted lines denote the 95 % significance levels range of the average values of the multi-model ensemble. The basin number is consistent with that given in Figure 1.

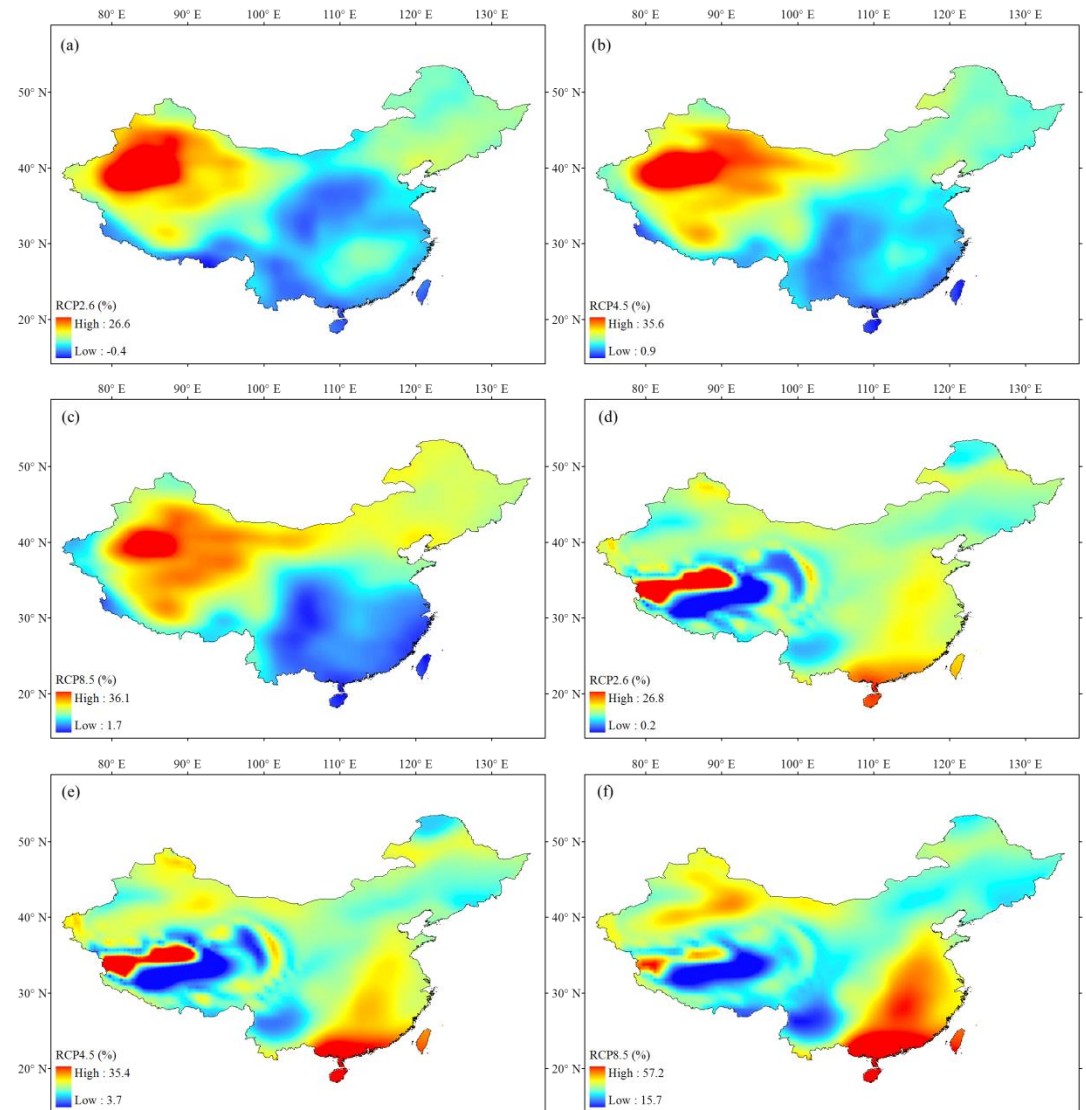


**Figure 10.** The CMIP5 multi-model ensemble median relative change (%) in mean annual *P* under (a)

RCP2.6, (b) RCP4.5, and (c) RCP8.5 scenarios and in mean annual *PET* under (d) RCP2.6, (e) RCP4.5,

and (f) RCP8.5 scenarios in China for the period 2071–2100 (relative to the baseline 1971–2000).







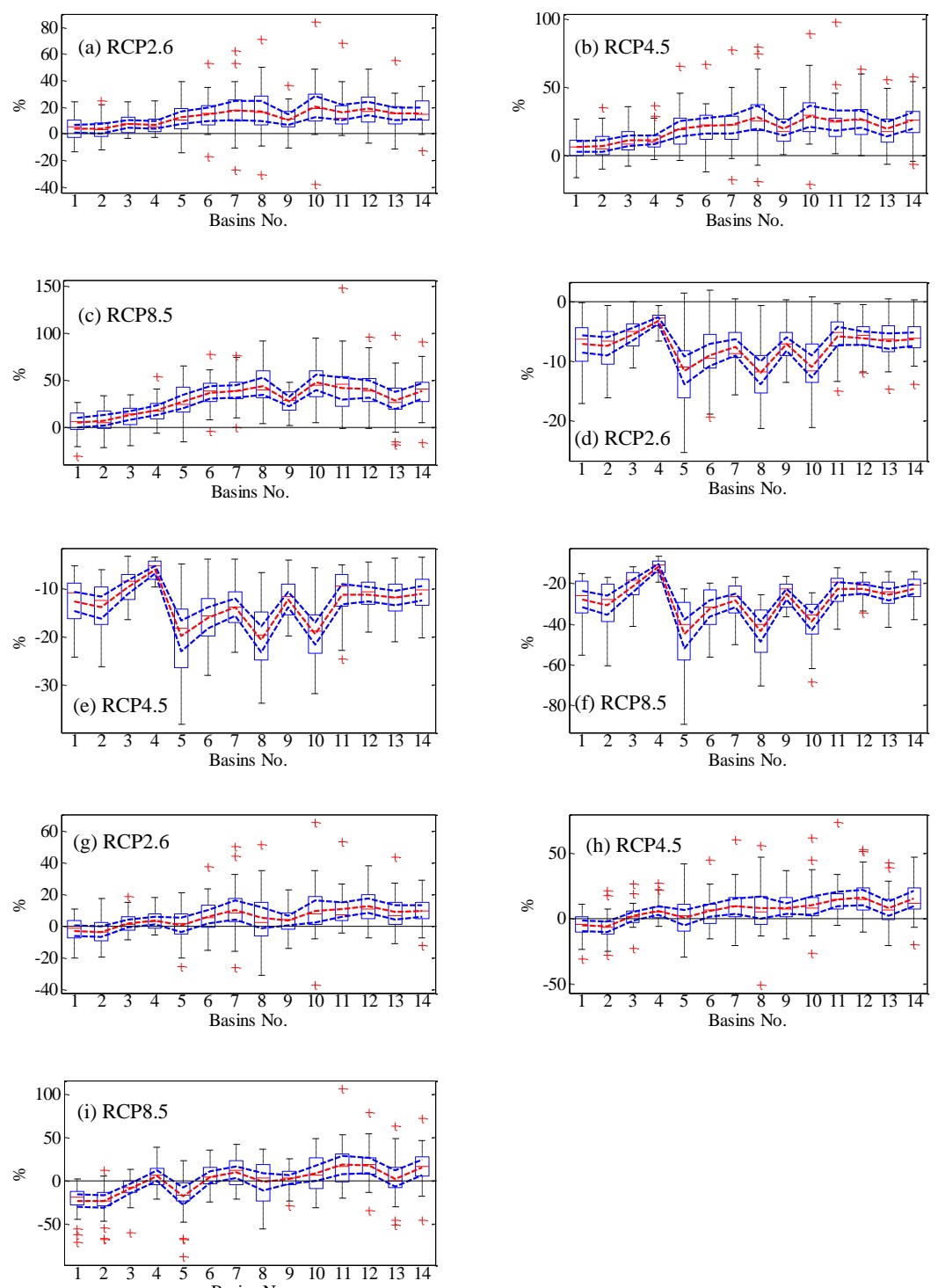


**Figure 11**. Box plots of relative change (%) in the contributions of annual *P* to *R* under (a) RCP2.6, (b) RCP4.5, and (c) RCP8.5 scenarios, in the contributions of annual *PET* to *R* under (d) RCP2.6, (e) RCP4.5, and (f) RCP8.5 scenarios, and in the contributions of climate to *R* under (g) RCP2.6, (h) RCP4.5, and (i) RCP8.5 scenarios calculated from 28 CMIP5 models in 14 basins for the period 2071– 2100 (relative to the baseline 1971–2000). The boxes denote the interquartile model spread (range between the 25th and 75th quantiles) with the horizontal line indicating the ensemble median and the whiskers showing the extreme range of the 28 CMIP5 model simulations. Red dotted lines denote the

average values of the multi-model ensemble. Blue dotted lines denote the 95% significance levels range
of the average values of the multi-model ensemble. The basin number is consistent with that given in
Figure 1.






















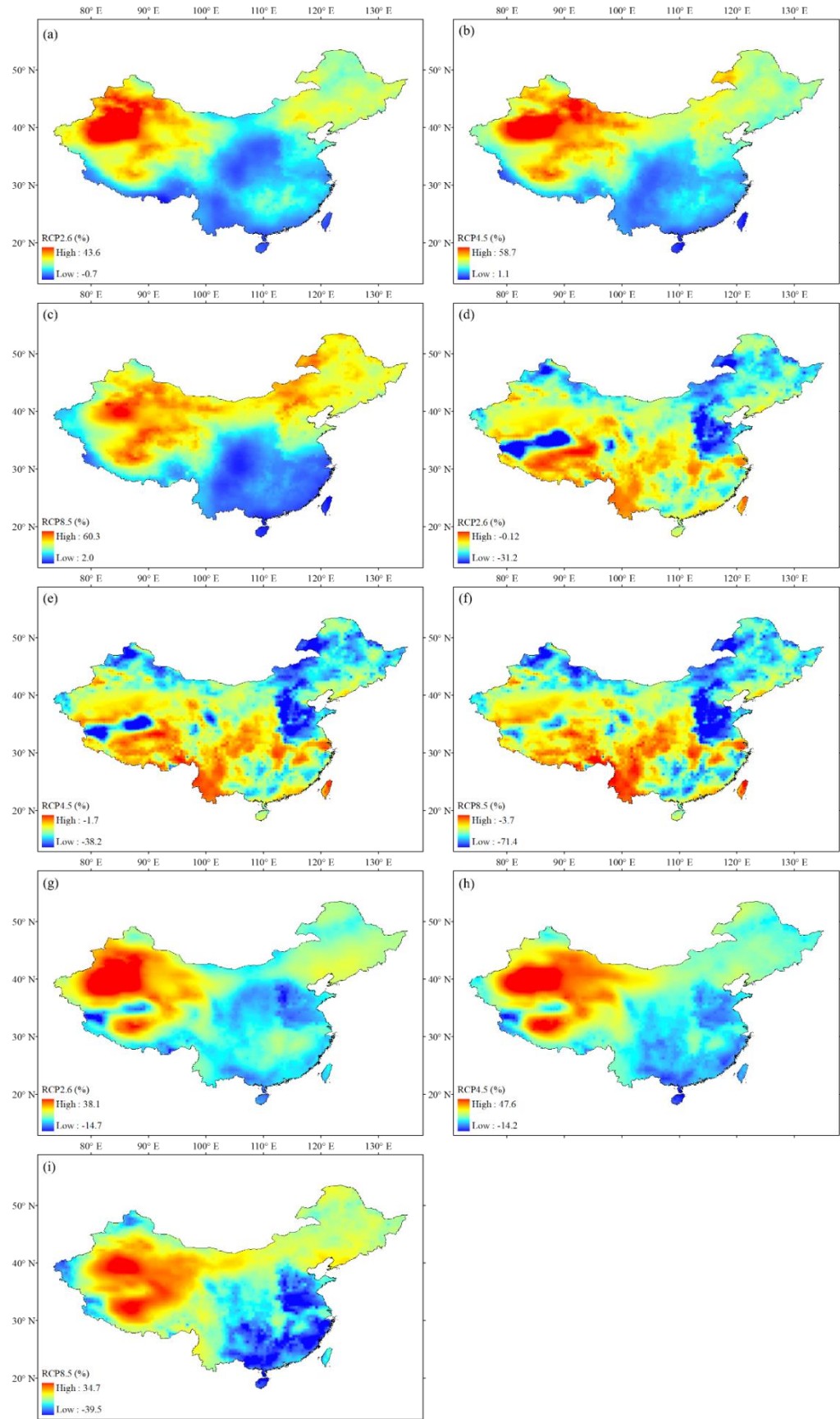


**Figure 12**. The CMIP5 multi-model ensemble median relative change (%) in the contributions of

annual *P* to *R* under (a) RCP2.6, (b) RCP4.5, and (c) RCP8.5 scenarios, in the contributions of annual

*PET* to *R* under (d) RCP2.6, (e) RCP4.5, and (f) RCP8.5 scenarios, and in the contributions of climate

to $R$ under (g) RCP2.6, (h) RCP4.5, and (i) RCP8.5 scenarios in China for the period 2071–2100
(relative to the baseline 1971–2000).























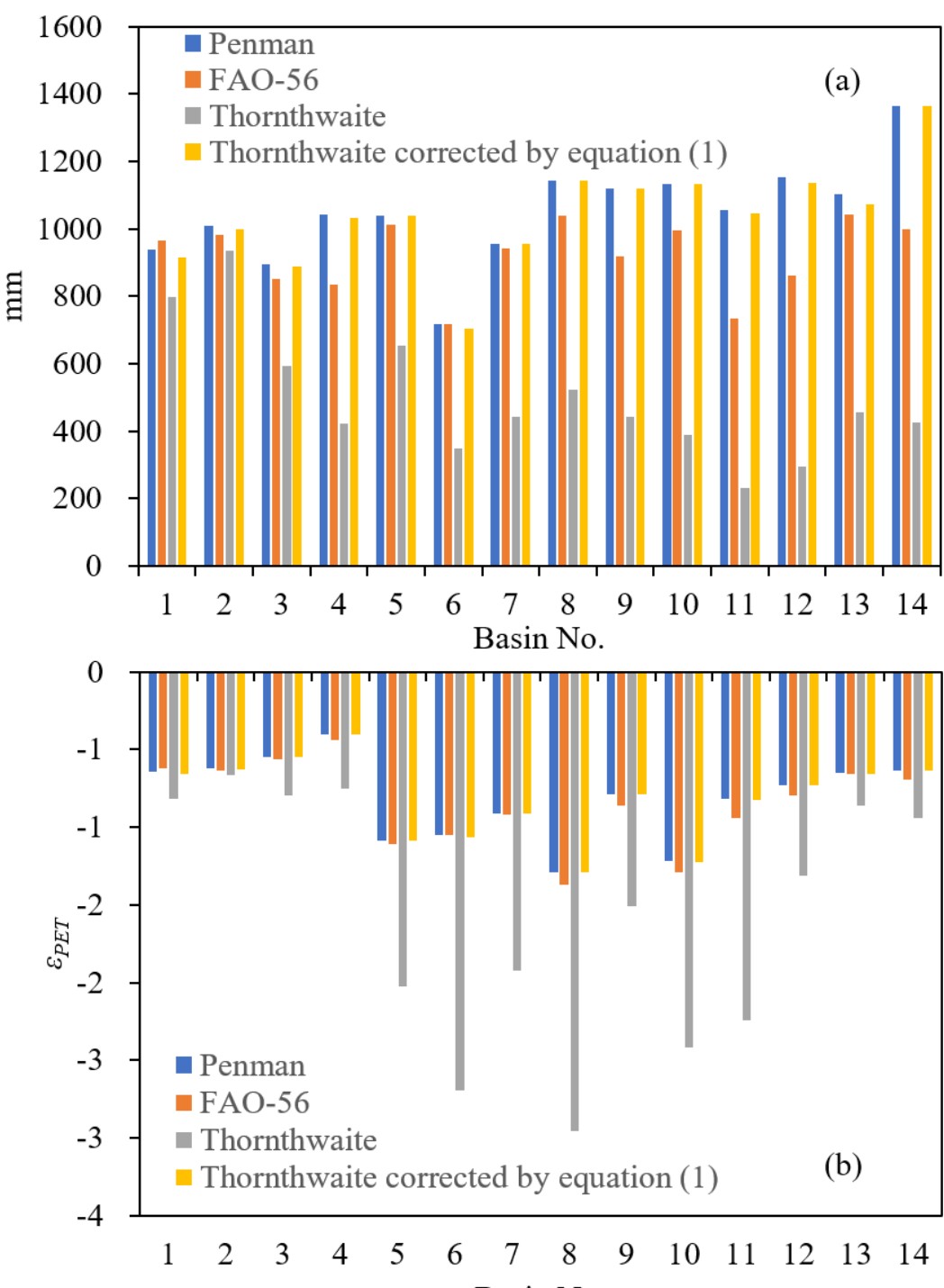


**Figure 13**. (a) Mean annual *PET* calculated from the four methods for the 14 river basins of China

during the period 1960–2008. (b) *PET* elasticity calculated from Equation (2) based on the four *PET*

data for the 14 river basins of China during the period 1960–2008. The basin number is consistent with

that given in Figure 1.


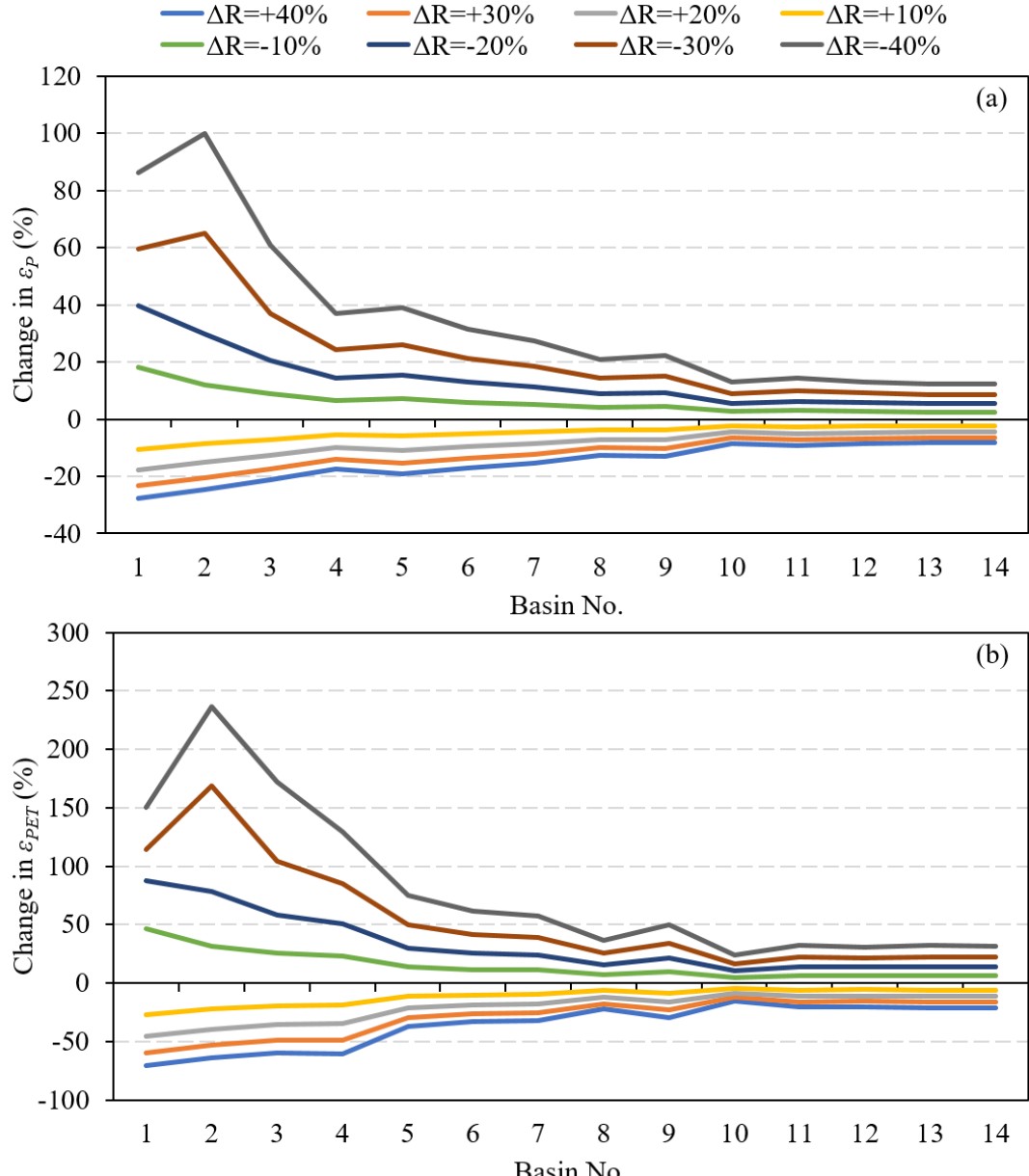


**Figure 14**. Comparison of changes in (a) *P* elasticity and (b) *PET* elasticity in response to changes in *R*
for the 14 river basins of China. The basin number is consistent with that given in Figure 1.