# Peer review of "Responses of runoff to historical and future climate variability over China"

_Hydrology and Earth System Sciences, 2017_

## Referee Comment (RC1) · Anonymous Referee #1 · 27 Apr 2017

This paper describes the projected effects of climate change on runoff and water availability in China using a framework based on runoff elasticity. In general the paper is well written and of sufficiently wide geographical scope to be interesting to a broad readership, but several key assumptions in the methodology, which are neither documented nor discussed, preclude a recommendation to publish without major revisions. These are:

1. More information on the parameters used in the hydrological modeling is necessary, especially those used with VIC to calculate runoff. These assumptions lie at the heart of the elasticities calculated, which will be heavily influenced by the structure and parametrisation of that model. [Section 2.1 Line 5].

2. There is some discussion of uncertainty in Section 4.3 but it is very general and not
quantitative. In particular, the detailed choice of which formulation of the Budyko model used to compute elasticities is investigated but neither the runoff model nor the PET equation are examined in this regard.

3. PET is calculated using the Thornthwaite method, which is a surprise since with the data available there is information to justify the use of more physically accurate PET calculations. Justification for the use of the temperature-based Thornthwaite method is required, especially given that it may oversimplify (and artificially constrain) the results of the Budyko calculation which features subsequently. [P5 line 5]
* * *

---

## Referee Comment (RC2) · Anonymous Referee #2 · 31 Jul 2017

This paper applies Budyko's concept of 'climate elasticity' in the response of runoff to changes in precipitation, potential evapotranspiration and catchment properties to projections of climate change from an ensemble of general circulation model projections. The authors use this to assess the robustness of projections of changes in future due to climate change in different regions of China.

Climate elasticity concept seems quite neat for the question of responses to climate change (separating P and PET drivers, and also with the potential for accounting for other drivers via the catchment properties) and in my opinion the authors have applied this appropriately to the specific question of responses to an ensemble of climate change projections. I would however advise more care in the interpretation, as these should not be taken as actual predictions of the future (which the language used sometimes suggests that there are). There are 3 reasons for this:

(1) While the use of the multi-model ensemble probably is a good, well-established way to explore a number of possible outcomes, the ensemble is not designed to be probabilistic, ie: it is not intended to give an indication of likelihoods. It is an 'ensemble of opportunity', using all models that happened to be available in the community, and the levels of skill for regional climate change in China will vary somewhat arbitrarily. The models themselves have not been specifically chosen or varied in order to systematically explore regional climate changes. Likelihood statements generally require further backing-up with understanding of model performance and the simulated climate processes in the region in question. Therefore I would encourage the authors to avoid terms such as "climate change will likely cause an obvious increase (decrease) of R" – the simulations are not intended to give guidance on likelihoods. (2) It is also not clear to me whether the catchment properties term includes plant stomatal responses to CO2. (It could do in theory). Two recent papers (Milly and Dunne, 2016, Nature Climate Change, and Swann et al, 2016, PNAS) showed that projected runoff changes in the GCMs tend to show a greater increase or smaller decrease in runoff than many hydrological models, because the GCM land surface schemes tend to include this term whereas hydrological models do not. It is not clear whether the VIC model includes this here or not. (3) The method used here does not, I believe, include other drivers of hydrological change eg. Land cover change, groundwater and river water extraction, irrigation etc. I think that in theory the catchment properties quantity could account for this, but it has not been applied to this here. We cannot assume that climate change is the only driver of hydrological change, and hence the interpretation of the results should bear this in mind.

The authors do acknowledge some of these issues to some extent at the end of the paper, but this is after the earlier discussion which often uses language of prediction, which I think goes too far. I would suggest terms such as "Climate change is projected to cause an increase (decrease) in R…." Also I suggest the authors address the above

points in more detail, highlighting the limits to the interpretation of the CMIP5 ensemble in terms of likelihoods.

My other concern is why the authors chose to use the Thorthwaite method for PET. It is stated on page 14 line 4 that this is because there is a "lack of meteorological data (such as relative humidity) in the GCM data. This is not true – GCMs are meteorological models, and indeed some of the CMIP5 GCMs are used in slightly different variants for numerical weather prediction. A huge range of meteorological outputs is available, including RH – see here http://cmip-pcmdi.llnl.gov/cmip5/docs/standard_output.pdf

I recommend that the authors use the data portal http://cmip-pcmdi.llnl.gov/cmip5/data_description.html at PCMDI, who organised CMIP5. The Canadian Climate Centre webpage used by the authors only has a very limited number of variables.

---

## Author Comment (AC1) · 26 Aug 2017

Dear Anonymous Referee #1,

Thanks very much for your constructive comments concerning our manuscript entitled "Responses of runoff to historical and future climate variability over China" (Manuscript No.: hess-2017-98). Those comments are all valuable and very helpful for revising and improving our paper, as well as the important guiding significance to our researches. We have studied comments carefully and here replied each comment below.

**\*\*\*\*\*\*\*\*\*\*\*\*\*\*\*\*\*\*\*\*\*\*\*\*\*\*\*\*\*\*\*\*\*\*\*\*\*\*\*\*\*\*\*\*\*\*\*\*\*\*\*\*\*\*\*\*\*\*\*\*\*\*\*\*\*\*\*\*\*\*\*\*\*\*\*\*\*\*\*\*\***

**Comments from Anonymous Referee #1:**

This paper describes the projected effects of climate change on runoff and water availability in China using a framework based on runoff elasticity. In general the paper is well written and of sufficiently wide geographical scope to be interesting to a broad readership, but several key assumptions in the methodology, which are neither documented nor discussed, preclude a recommendation to publish without major revisions. These are:

**1.** More information on the parameters used in the hydrological modeling is necessary, especially those used with VIC to calculate runoff. These assumptions lie at the heart of the elasticities calculated, which will be heavily influenced by the structure and parametrisation of that model. [Section 2.1 Line 5]

**Response:** Thank you very much for your nice comments. In our study, the Budyko framework with an empirical parameter was used to calculate the climate elasticity of runoff, and this method has been proven to be robust to the calculation of climate elasticity (Yang et al., 2014). For the VIC model used for the calculation of runoff, the parameters include: the infiltration parameter $b$, the second and third soil layer depths ($d_2$ and $d_3$), and the three parameters in the base flow scheme. According to Zhang et al (2014), the VIC model was calibrated in the 11 major basins over China based on the best meteorological forcing data (derived by 756 meteorological stations over China). The model parameters were estimated by using an optimization algorithm of the multi-objective complex evolution of the University of Arizona (MOCOM-UA). The results indicated that the simulated runoff matches reasonably well with the observations at both seasonal and monthly timescales, with the Nash–Sutcliffe efficiency above 0.8. Our study conducted the analysis of climate elasticity of runoff at the long-term scale using annual data of runoff, which would be more accurate than that at the monthly scale. Therefore, the simulated annual runoff by the VIC model would show little influence on the calculation of elasticity. According to your good comments, we have added more information on the parameters in the VIC modeling in the revised manuscript.

**Reference:**

Yang, H., Qi, J., Xu, X., Yang, D., and Lv, H.: The regional variation in climate elasticity and climate contribution to runoff across China. J. hydrol., 517, 607-616, 2014.

Zhang, X., Tang, Q., Pan, M., Tang, Y.: A Long-Term Land Surface Hydrologic Fluxes and States Dataset for China, J. Hydrometeor., 15, 2067–2084, doi: 10.1175/JHM-D-13-0170.1, 2014.

**2.** There is some discussion of uncertainty in Section 4.3 but it is very general and not quantitative. In particular, the detailed choice of which formulation of the Budyko model used to compute elasticities is investigated but neither the runoff model nor the *PET* equation are examined in this regard.

**Response:** Thank you very much for your nice comments. We agree with you that the discussion section is lack of quantitative analysis, especially for the examination of the estimation of runoff or *PET*. The runoff simulated by the VIC model has been proven to be accurate at the long-term scale (Zhang et al., 2014). In our original version (i.e. initial submission), the *PET* of the 28 GCMs for the baseline 1971–2000 and the future period 2071–2100 was estimated by the Thornthwaite method. We noted that the Thornthwaite method is solely based on monthly temperature, which may tend to underestimate *PET* in the arid areas and overestimate *PET* in the humid areas. Therefore, we used a multiplicative correction for *PET* bias correction of the 28 GCMs:

$$PET_{cor,GCM,i} = PET_{Th,GCM,i} \times \frac{\overline{PET}_{Pen,obs,i}}{\overline{PET}_{Th,obs,i}} \tag{1}$$

where $PET_{Th,GCM,i}$ and $PET_{cor,GCM,i}$ are annual *PET* from the Thornthwaite method and the bias-corrected annual *PET*, respectively, for the *i*th grid point of the GCM data. $\overline{PET}_{Pen,obs,i}$ and $\overline{PET}_{Th,obs,i}$ are the 49-year averages of *PET* calculated from the Penman and Thornthwaite methods, respectively, for the *i*th grid point for the period 1960–2008.

According to your good suggestions, we compared four different *PET* calculation equations (i.e., the Penman method, the Thornthwaite method, the FAO-56 Penman–Monteith method, and the Thornthwaite method corrected by equation (1)) over the 14 river basins of China, and conducted a quantitative analysis of the impacts of the *PET* calculations on the *PET* elasticity calculations (as shown in Figure R1 below). The results showed that the mean annual *PET* by the Penman method, the FAO-56 Penman–Monteith method, and the Thornthwaite method corrected by equation (1) are quite consistent, and the *PET* elasticity calculations from these three methods give very similar results in all 14 basins. In summary, our study suggests that the estimation of *PET* elasticity is robust to the *PET* estimated from the Penman method, the FAO-56 method, and the Thornthwaite method corrected by equation (1), but is not robust to the Thornthwaite method.

[Figure]

Figure R1. (a) Mean annual *PET* calculated from the four methods for the 14 river basins of China during the period 1960–2008. (b) *PET* elasticity calculated based on the four *PET* data for the 14 river basins of China during the period 1960–2008. The basin number is as follows: 1, Southeast Drainage; 2, Pearl River; 3, Yangtze River; 4, Southwest Drainage; 5, Huaihe River; 6, Heilongjiang River; 7, Liaohe River; 8, Haihe River; 9, Yellow River; 10, Inner Mongolia River; 11, Qiangtang River; 12, Qinghai River; 13, Xinjiang River, 14, Hexi River.

**Reference:**

Zhang, X., Tang, Q., Pan, M., Tang, Y.: A Long-Term Land Surface Hydrologic Fluxes and States Dataset for China, J. Hydrometeor., 15, 2067–2084, doi: 10.1175/JHM-D-13-0170.1, 2014.

**3.** *PET* is calculated using the Thornthwaite method, which is a surprise since with the data available there is information to justify the use of more physically accurate *PET* calculations. Justification for the use of the temperature-based Thornthwaite method is required, especially given that it may oversimplify (and artificially constrain) the results of the Budyko calculation which features subsequently. [P5 line 5]

**Response:** Thank you very much for your nice comments. In the original version (i.e. initial submitted manuscript), the *PET* data used for the calculation of climate elasticity are derived from the CRU TS3.22 dataset as produced by the Climatic Research Unit (CRU) at the University of East Anglia (Harris et al., 2014). In this dataset, the *PET* is calculated from the FAO Penman-Monteith method. In contrast, the *PET* of the 28 GCMs is estimated by the Thornthwaite method. We fully agree with you that the temperature-based Thornthwaite method is lack of physical basis, and it is necessary to justify the use of the temperature-based Thornthwaite method and the use of more physically *PET* calculation methods.

In the revised manuscript, we used a more physically *PET* data that estimated by the Penman method (during the period 1960–2008 provided by the Hydroclimatology Group of Princeton University) to calculate the climate elasticity (i.e. *PET* elasticity) over China instead of the *PET* data from the FAO Penman-Monteith method. The related results of the study and some figures and tables have been updated in the revised manuscript. We believe the new climate elasticity coefficients would be more accurate compared with that in the original version. Meanwhile, the *PET* of GCMs calculated by the Thornthwaite method was corrected by the equation (1) above. We compared the corrected annual *PET* with the *PET* calculated from the Penman method at both basin and grid scales (Figure R2). The results indicated the Thornthwaite method corrected by the equation (1) significantly improves the accuracy of *PET* and can be acceptable for the *PET* calculation of the 28 GCMs. In future work, we are going to compute the Penman *PET* using the meteorological data from the CMIP5 output and make a comparative analysis to fully understand the *PET* calculation uncertainties in the projections of climate change.

[Figure]

Figure R2. Comparison of annual *PET* calculated from the Penman method and the Thornthwaite method corrected by Equation (1) during the period 1960–2008 for (a) the 14 river basins and (b) all 0.5° grid points over China.

**Reference:**
Harris, I., Jones, P. D., Osborna, T. J., Lister, D. H.: Updated high-resolution grids of monthly climatic observations–the CRU TS3.10 Dataset, Int. J. Climatol., 34(3), 623–642, 2014.

---

## Author Comment (AC2) · 26 Aug 2017

Dear Anonymous Referee #2,

Thanks very much for your constructive comments concerning our manuscript entitled "Responses of runoff to historical and future climate variability over China" (Manuscript No.: hess-2017-98). Those comments are all valuable and very helpful for revising and improving our paper, as well as the important guiding significance to our researches. We have studied comments carefully and here replied each comment below.

**\*\*\*\*\*\*\*\*\*\*\*\*\*\*\*\*\*\*\*\*\*\*\*\*\*\*\*\*\*\*\*\*\*\*\*\*\*\*\*\*\*\*\*\*\*\*\*\*\*\*\*\*\*\*\*\*\*\*\*\*\*\*\*\*\*\*\*\*\*\*\*\*\*\*\*\***

**Comments from Anonymous Referee #2:**

This paper applies Budyko's concept of 'climate elasticity' in the response of runoff to changes in precipitation, potential evapotranspiration and catchment properties to projections of climate change from an ensemble of general circulation model projections. The authors use this to assess the robustness of projections of changes in future due to climate change in different regions of China.

Climate elasticity concept seems quite neat for the question of responses to climate change (separating P and PET drivers, and also with the potential for accounting for other drivers via the catchment properties) and in my opinion the authors have applied this appropriately to the specific question of responses to an ensemble of climate change projections. I would however advise more care in the interpretation, as these should not be taken as actual predictions of the future (which the language used some- times suggests that there are). There are 3 reasons for this:

**1.** (1) While the use of the multi-model ensemble probably is a good, well-established way to explore a number of possible outcomes, the ensemble is not designed to be probabilistic, ie: it is not intended to give an indication of likelihoods. It is an 'ensemble of opportunity', using all models that happened to be available in the community, and the levels of skill for regional climate change in China will vary somewhat arbitrarily. The models themselves have not been specifically chosen or varied in order to systematically explore regional climate changes. Likelihood statements generally require further backing-up with understanding of model performance and the simulated climate processes in the region in question. Therefore I would encourage the authors to avoid terms such as "climate change will likely cause an obvious increase (decrease) of R" – the simulations are not intended to give guidance on likelihoods. (2) It is also not clear to me whether the catchment properties term includes plant stomatal responses to CO2. (It could do in theory). Two recent papers (Milly and Dunne, 2016, Nature Climate Change, and Swann et al, 2016, PNAS) showed that projected runoff changes in the GCMs tend to show a greater increase or smaller decrease in runoff than many hydrological models, because the GCM land surface schemes tend to include this term whereas hydrological models do not. It is not clear whether the VIC model includes this here or not. (3) The method used here does not, I believe, include other drivers of hydrological change eg. Land cover change, groundwater and river water extraction, irrigation etc. I think that in theory the catchment properties quantity could account for this, but it has not been applied to this here. We cannot assume that climate change is the only driver of hydrological change, and hence the interpretation of the results should bear this in mind.

Response: Thank you very much for your nice comments. For the question 1, we quite agree with

your points that the multi-model ensemble is not designed to be probabilistic and is not intended to give an indication of likelihoods. Therefore, likelihood statements, which generally require further backing-up with understanding of model performance and the simulated climate processes, are not appropriate here. According to your good suggestions, we have changed the statements of some sentences in the revised manuscript to avoid term such as 'climate change will likely cause an obvious increase (decrease) of $R$' (changed to 'climate change is projected to cause an increase (decrease) in $R$').

For the question 2, thank you for providing these two very nice references (Milly and Dunne, 2016, Swann et al, 2016), which showed a very important information that the plant responses to increasing $CO_2$ tend to increase the amount of water on land, leading to a greater increase in runoff. We note that the VIC model used for the calculation of runoff does not include the schemes of the plant stomatal responses to $CO_2$. Therefore, under high $CO_2$ condition, neglecting the plant stomatal responses to $CO_2$ would lead to the underestimation of runoff in the hydrological model. According to your good comments, we made a discussion on this point to highlight the importance of the plant stomatal responses to $CO_2$ in the assessment of hydrological impacts of climate change. In addition, the empirical parameter in the Budyko equations well accounts for the effects of catchment properties (e.g. land surface characteristics, the average slope, and vegetation type) on the water-energy balance. Therefore, the catchment properties term could include plant stomatal responses to $CO_2$ in theory. This is a very nice suggestion for us to try to characterize the plant stomatal responses to $CO_2$ using the catchment properties term in the future work, especially under high $CO_2$ condition.

For the question 3, we quite agree with your comments that there are also other drivers of hydrological change in addition to climate change. Our method only considers the hydrological change due to climate change but neglects the effects of the variability of catchment properties (e.g., land cover change, groundwater and river water extraction, urbanization, irrigation, etc.) on the hydrology. According to your good comments, we made a discussion on the other driver (catchment properties) of hydrological change for the interpretation of the results in the revised manuscript.

**2.** The authors do acknowledge some of these issues to some extent at the end of the paper, but this is after the earlier discussion which often uses language of prediction, which I think goes too far. I would suggest terms such as "Climate change is projected to cause an increase (decrease) in R$\because$." Also I suggest the authors address the above points in more detail, highlighting the limits to the interpretation of the CMIP5 ensemble in terms of likelihoods.

**Response:** Thank you very much for your nice comments. According to your good suggestions, we have changed the sentence "climate change will likely cause an obvious increase (decrease) of $R$…" to "climate change is projected to cause an increase (decrease) in $R$…". We also addressed the above points in more detail in the revised manuscript to highlight the limits to the interpretation of the CMIP5 ensemble in terms of likelihoods.

**3.** My other concern is why the authors chose to use the Thorthwaite method for PET. It is stated

on page 14 line 4 that this is because there is a "lack of meteorological data (such as relative humidity) in the GCM data. This is not true – GCMs are meteorological models, and indeed some of the CMIP5 GCMs are used in slightly different variants for numerical weather prediction. A huge range of meteorological outputs is available, including RH – see here http://cmip-pcmdi.llnl.gov/cmip5/docs/standard_output.pdf.

I recommend that the authors use the data portal http://cmippcmdi.llnl.gov/cmip5/data_description.html at PCMDI, who organised CMIP5. The Canadian Climate Centre webpage used by the authors only has a very limited number of variables.

**Response:** Thank you very much for your nice comments. In the original version (i.e. initial submitted manuscript), the *PET* of GCM for the baseline 1971–2000 and the future period 2071–2100 is estimated by the Thornthwaite method. We noted that the temperature-based Thornthwaite method is lack of physical basis, and it is necessary to justify the use of the Thornthwaite method and the use of more physically *PET* calculation methods. Thank you very much for informing us that the meteorological data used for the calculation of *PET* is available from the CMIP5 output http://cmippcmdi.llnl.gov/cmip5/data_description.html at PCMDI. Indeed, there is a huge range of meteorological outputs (including RH) from the CMIP5 models, which are enough for the calculation of *PET* by the Penman method. However, due to large amounts of data needed to be processed (including (1) download the 28 GCMs meteorological data, (2) statistical downscaling of the 28 GCMs meteorological data over China, (3) bias correction of the 28 GCMs meteorological data, (4) calculations of *PET* for the 28 GCMs, and (5) bias correction of *PET* for the 28 GCMs), it is difficult for us to complete it in a short period.

However, we tried our best to correct the *PET* of GCMs calculated by the Thornthwaite method, and made a detailed comparison of the corrected *PET* method with other *PET* calculation methods to justify the use of *PET* calculation of the GCMs. In particular, there are three main changes for the *PET* calculations in the revised manuscript, which are as follows:

(1) We used a more physically *PET* data that estimated by the Penman equation (data during the period 1960–2008 provided by the Hydroclimatology Group of Princeton University) to calculate the climate elasticity (i.e. *PET* elasticity) over China instead of the *PET* data from the FAO Penman-Monteith method. We believe the climate elasticity would be more accurate in the revised manuscript than in the original version.

(2) We used a multiplicative correction method to correct the *PET* data of GCMs calculated from the Thornthwaite method as follows:

$$PET_{cor,GCM,i} = PET_{Th,GCM,i} \times \frac{\overline{PET}_{Pen,obs,i}}{\overline{PET}_{Th,obs,i}} \tag{1}$$

where $PET_{Th,GCM,i}$ and $PET_{cor,GCM,i}$ are annual *PET* from the Thornthwaite method and the

bias-corrected annual *PET*, respectively, for the *i*th grid point of the GCM data. $\overline{PET}_{Pen,obs,i}$ and

$\overline{PET}_{Th,obs,i}$ are the 49-year averages of *PET* calculated from the Penman method and Thornthwaite method, respectively, for the *i*th grid point for the period 1960–2008.

Based on the monthly data of temperature covering the period 1960–2008 provided by the Climatic Research Unit (CRU), the *PET* was calculated by the Thornthwaite method and then corrected by the equation (1) to test the applicability of the multiplicative correction method. The results indicated that the corrected annual *PET* shows a good agreement with that calculated by the Penman method (as shown in Figure R1). These two methods are quite consistent at both basin and grid scales, suggesting that the equation (1) above is acceptable for the bias correction of *PET* of the GCMs.

(3) We compared the four *PET* calculation methods (i.e., the Penman method, the Thornthwaite method, the FAO-56 Penman–Monteith method, and the Thornthwaite method corrected by the equation (1)) to test the robustness of the *PET* elasticity result subject to *PET* uncertainties. The results indicated that the mean annual *PET* by the Penman method, the FAO-56 Penman–Monteith method, and the Thornthwaite method corrected by the equation (1) are quite consistent, and the *PET* elasticity calculations from these three methods give very similar results in all 14 basins (as shown in Figure R2). That is to say, the Thornthwaite method corrected by the equation (1) significantly improves the accuracy of *PET* and can be acceptable for the *PET* calculation of the GCMs.

In the future work, we are going to calculate the Penman *PET* using the meteorological data from the CMIP5 output and make a comparative analysis to fully understand the *PET* calculation uncertainties in the projections of climate change.

[Figure]

Figure R1. Comparison of annual *PET* calculated from the Penman method and the Thornthwaite method corrected by Equation (1) during the period 1960–2008 for (a) the 14 river basins and (b) all 0.5° grid points over China.

[Figure]

Figure R2. (a) Mean annual *PET* calculated from the four methods for the 14 river basins of China during the period 1960–2008. (b) *PET* elasticity calculated based on the four *PET* data for the 14 river basins of China during the period 1960–2008. The basin number is as follows: 1, Southeast Drainage; 2, Pearl River; 3, Yangtze River; 4, Southwest Drainage; 5, Huaihe River; 6, Heilongjiang River; 7, Liaohe River; 8, Haihe River; 9, Yellow River; 10, Inner Mongolia River; 11, Qiangtang River; 12, Qinghai River; 13, Xinjiang River, 14, Hexi River.

---

## Author Response (AR1)

Dear Editor and Reviewers:

On behalf of all the contributing authors, I would like to express our sincere appreciations of your letter and reviewers' constructive comments concerning our article entitled "Responses of runoff to historical and future climate variability over China" (Manuscript No.: **hess-2017-98**). Those comments are all valuable and very helpful for revising and improving our paper, as well as the important guiding significance to our researches. We have studied comments carefully and have made correction which we hope meet with approval. In this revised version, changes to our manuscript were all highlighted within the document by using red colored text. Point-by-point responses to the nice editor and two nice reviewers are listed below this letter.

**\*\*\*\*\*\*\*\*\*\*\*\*\*\*\*\*\*\*\*\*\*\*\*\*\*\*\*\*\*\*\*\*\*\*\*\*\*\*\*\*\*\*\*\*\*\*\*\*\*\*\*\*\*\*\*\*\*\*\*\*\*\*\*\*\*\*\*\*\*\***

**Editor comments:**

I think the authors have responded to some critical comments well. I do class this paper as requiring major corrections to answer some of these points and that is in agreement with both reviewers. Please can the authors submit a revised manuscript that will be further reviewed by both the current reviewers. To add to the authors comments I note the following that I would like to see more developed:

Reviewer 1, comment 1: I think there needs to be a more quantified answer to whether or not the modelled results may impact the elasticities calculated. I'm not sure I agree that a calibration scheme that is focused on high flows and having some uncertainty (as all models do) means that 'annual' outputs are more accurate. I can imagine the extent of that is catchment response dependent... So ensure this effect is proved by the manuscripts analyses. Please note that the reviews are asking for a treatment of the uncertainties in the modelling and this is not all about different PET calculations in my view....

Also I will want to make sure then authors identify and are seen to be dealing with the issues of hydrological model simulations to future climates and how 'valid' there modelling system is for achieving this (from the GCM's downwards through the hydrological cascade).

**Response:** Thank you very much for your nice comments. We quite agree with you that there needs to be a more quantified answer to whether or not the modelled results may impact the elasticities calculated. According to your good suggestion, we further made a sensitivity analysis on the changes in $P$ elasticity and $PET$ elasticity in response to changes in runoff ($R$) for the 14 river basins in China (As shown in Figure 14 in the revised MS). We found that the sensitivity of climate (i.e., $P$ and $PET$) elasticity to $R$ varies considerably between basins and tends to be larger in more humid basins. Moreover, $PET$ elasticity is more sensitive to changes in $R$ compared with $P$ elasticity for all 14 basins. As indicated by Zhang et al. (2014), the $R$ is realistically estimated for most of the basins (especially for humid basins) in China with a small relative error, but there is a large relative error for few arid basins in western China due to the lack of meteorological observations. Therefore, these errors in simulated $R$ of the VIC model may result in uncertainties in elasticity calculation, particularly in western China. For more information please see lines 396-403 in the revised MS.

To address the issues of hydrological simulations to future climate change, our study firstly calculated the climate (i.e., $P$ and $PET$) elasticity of $R$ (i.e., per change in $R$ due to per change in $P$ and $PET$) over China by the Budyko-based elasticity method, based on the land surface data from the VIC model (Zhang et al., 2014). Then we projected changes in climate (i.e., changes in *P* and *PET*) over the 0.5 degree grids of China during the period 2071-2100 in RCPs 4.5 and 8.5 compared with the baseline period (1971-2000), by using the downscaling results of the 28 GCMs. By neglecting the catchment properties elasticity, the projected changes in *P* and *PET* over China from the 28 GCMs were taken into equation (4) (as shown in the revised MS) to project future hydrological changes (i.e. projected changes in *R*) due to climate change during the period 2071-2100 in RCPs 4.5 and 8.5. We think climate elasticity concept is neat for the issues of hydrological responses to an ensemble of climate change projections.

**Reference:**

Zhang, X., Tang, Q., Pan, M., Tang, Y.: A Long-Term Land Surface Hydrologic Fluxes and States Dataset for China, J. Hydrometeor., 15, 2067–2084, doi: 10.1175/JHM-D-13-0170.1, 2014.

**\*\*\*\*\*\*\*\*\*\*\*\*\*\*\*\*\*\*\*\*\*\*\*\*\*\*\*\*\*\*\*\*\*\*\*\*\*\*\*\*\*\*\*\*\*\*\*\*\*\*\*\*\*\*\*\*\*\*\*\*\*\*\*\*\*\*\*\*\***

**Anonymous Referee #1:**

This paper describes the projected effects of climate change on runoff and water availability in China using a framework based on runoff elasticity. In general the paper is well written and of sufficiently wide geographical scope to be interesting to a broad readership, but several key assumptions in the methodology, which are neither documented nor discussed, preclude a recommendation to publish without major revisions. These are:

**1.** More information on the parameters used in the hydrological modeling is necessary, especially those used with VIC to calculate runoff. These assumptions lie at the heart of the elasticities calculated, which will be heavily influenced by the structure and parametrisation of that model. [Section 2.1 Line 5]

**Response:** Thank you very much for your nice comments. In our study, the Budyko framework with an empirical parameter was used to calculate climate elasticity of runoff (*R*), and this method has been proven to be robust to the calculation of climate elasticity (Yang et al., 2014). For the VIC model used for the calculation of runoff, the parameters include: the infiltration parameter *b*, the second and third soil layer depths ($d_2$ and $d_3$), and the three parameters in the base flow scheme. According to Zhang et al (2014), the VIC model was calibrated in the 11 major basins over China based on the best meteorological forcing data (derived by 756 meteorological stations over China). The model parameters were estimated by using an optimization algorithm of the multi-objective complex evolution of the University of Arizona (MOCOM-UA). According to your good comments, we have added more information on the parameters in the VIC modeling in the revised MS. For detailed information please see lines 89-93 in the revised MS.

In addition, we made a sensitivity analysis on the changes in *P* elasticity and *PET* elasticity in response to changes in *R* for the 14 river basins in China (As shown in Figure 14 in the revised MS). We found that the sensitivity of climate (i.e., *P* and *PET*) elasticity to *R* varies considerably between basins and tends to be larger in more humid basins. Moreover, *PET* elasticity is more sensitive to changes in *R* compared with *P* elasticity. As indicated by Zhang et al. (2014), the *R* is realistically estimated for most of the basins (especially for humid basins) in China with a small relative error, but there is a large relative error for few arid basins in western China due to the lack of meteorological observations. Therefore, the results suggest that the errors in simulated *R* of the VIC model may result in uncertainties in elasticity calculation, particularly in western China. For more information please see lines 396-403 in the revised MS.

**Reference:**

Yang, H., Qi, J., Xu, X., Yang, D., and Lv, H.: The regional variation in climate elasticity and climate contribution to runoff across China. J. hydrol., 517, 607-616, 2014.

Zhang, X., Tang, Q., Pan, M., Tang, Y.: A Long-Term Land Surface Hydrologic Fluxes and States Dataset for China, J.

Hydrometeor., 15, 2067–2084, doi: 10.1175/JHM-D-13-0170.1, 2014.

**2.** There is some discussion of uncertainty in Section 4.3 but it is very general and not quantitative. In particular, the detailed choice of which formulation of the Budyko model used to compute elasticities is investigated but neither the runoff model nor the *PET* equation are examined in this regard.

**Response:** Thank you very much for your nice comments. We agree with you that the discussion section is lack of quantitative analysis, especially for the examination of the estimation of runoff or *PET*. In our original version (i.e. initial submission), the *PET* of the 28 GCMs for the baseline 1971–2000 and the future period 2071–2100 was estimated by the Thornthwaite method. We noted that the Thornthwaite method is solely based on monthly temperature, which may tend to underestimate *PET* in the arid areas and overestimate *PET* in the humid areas. Therefore, we used a multiplicative correction for *PET* bias correction of the 28 GCMs (as shown in equation (1) in the revised MS).

According to your good suggestions, we compared four different *PET* calculation equations (i.e., the Penman method, the Thornthwaite method, the FAO-56 Penman–Monteith method, and the Thornthwaite method corrected by equation (1) in the revised MS) over the 14 river basins of China, and conducted a quantitative analysis of the impacts of the *PET* calculations on the *PET* elasticity calculations (as shown in Figure 13 in the revised MS). The results showed that the mean annual *PET* by the Penman method, the FAO-56 Penman–Monteith method, and the Thornthwaite method corrected by equation (1) are quite consistent, and the *PET* elasticity calculations from these three methods give very similar results in all 14 basins. In summary, our study suggests that the estimation of *PET* elasticity is robust to the *PET* estimated from the Penman method, the FAO-56 method, and the Thornthwaite method corrected by equation (1), but is not robust to the Thornthwaite method. For more information please see section 4.2 in the revised MS.

We also made a discussion on the comparison of changes in *P* elasticity and *PET* elasticity in response to changes in *R* for the 14 river basins in China (As shown in Figure 14 in the revised MS). It was found that the sensitivity of climate (i.e., *P* and *PET*) elasticity to *R* varies considerably between basins and tends to be larger in more humid basins. Moreover, *PET* elasticity is more sensitive to changes in *R* compared with *P* elasticity. As shown in Zhang et al. (2014), the *R* is realistically estimated for most of the basins (especially for humid basins) in China with a small relative error. However, there is a large relative error for few arid basins in western China due to the lack of meteorological observations. Therefore, our results suggest that the errors in simulated *R* of the VIC model may result in uncertainties in elasticity calculation, and this is particularly in western China. For more information please see lines 396-403 in the revised MS.

**\*\*\*\*\*\*\*\*\*\*\*\*\*\*\*\*\*\*\*\*\*\*\*\*\*\*\*\*\*\*\*\*\*\*\*\*\*\*\*\*\*\*\*\*\*\*\*\*\*\*\*\*\*\*\*\*\*\*\*\*\*\*\*\*\*\*\*\*\*\***

**Anonymous Referee #2:**

This paper applies Budyko's concept of 'climate elasticity' in the response of runoff to changes in precipitation, potential evapotranspiration and catchment properties to projections of climate change from an ensemble of general circulation model projections. The authors use this to assess the robustness of projections of changes in future due to climate change in different regions of China.

Climate elasticity concept seems quite neat for the question of responses to climate change (separating P and PET drivers, and also with the potential for accounting for other drivers via the catchment properties) and in my opinion the authors have applied this appropriately to the specific question of responses to an ensemble of climate change projections. I would however advise more care in the interpretation, as these should not be taken as actual predictions of the future (which the language used some- times suggests that there are). There are 3 reasons for this:
**1.** (1) While the use of the multi-model ensemble probably is a good, well-established way to explore a number of possible outcomes, the ensemble is not designed to be probabilistic, ie: it is not intended to give an indication of likelihoods. It is an 'ensemble of opportunity', using all models that happened to be available in the community, and the levels of skill for regional climate change in China will vary somewhat arbitrarily. The models themselves have not been specifically chosen or varied in order to systematically explore regional climate changes. Likelihood statements generally require further backing-up with understanding of model performance and the simulated climate processes in the region in question. Therefore I would encourage the authors to avoid terms such as "climate change will likely cause an obvious increase (decrease) of R" – the simulations are not intended to give guidance on likelihoods. (2) It is also not clear to me whether the catchment properties term includes plant stomatal responses to CO2. (It could do in theory). Two recent papers (Milly and Dunne, 2016, Nature Climate Change, and Swann et al, 2016, PNAS) showed that projected runoff changes in the GCMs tend to show a greater increase or smaller decrease in runoff than many hydrological models, because the GCM land surface schemes tend to include this term whereas hydrological models do not. It is not clear whether the VIC model includes this here or not. (3) The method used here does not, I believe, include other drivers of hydrological change eg. Land cover change, groundwater and river water extraction, irrigation etc. I think that in theory the catchment properties quantity could account for this, but it has not been applied to this here. We cannot assume that climate change is the only driver of hydrological change, and hence the interpretation of the results should bear this in mind.

Response: Thank you very much for your nice comments. For the question 1, we quite agree with your points that the multi-model ensemble is not designed to be probabilistic and is not intended to give an indication of likelihoods. Likelihood statements, which generally require further backing-up with understanding of model performance and the simulated climate processes, are not appropriate here. According to your good suggestions, we have changed the statements of some sentences to avoid term such as 'climate change will likely cause an obvious increase (decrease) of $R$' (changed to 'climate change is projected to cause an increase (decrease) in $R$'). For more information please see the red colored text in the revised MS.

For the question 2, thank you for providing these two very nice references (Milly and Dunne, 2016, Swann et al, 2016), which showed a very important information that the plant responses to increasing $CO_2$ tend to save more water on land, leading to a greater increase in runoff. We note that the VIC model used for the calculation of runoff does not include the schemes with the plant stomatal responses to $CO_2$. Therefore, under high $CO_2$ condition, neglecting the plant stomatal responses to $CO_2$ would lead to a underestimation of runoff in the hydrological model. According to your good comments, we made a discussion on this point to highlight the importance of the plant stomatal responses to $CO_2$ in the assessment of hydrological impacts of climate change. For more information please see lines 368-372 in the revised MS. In addition, the empirical parameter in the Budyko equations well accounts for the effects of catchment properties (e.g. land surface characteristics, the average slope, and vegetation type) on the water-energy balance. Therefore, the catchment properties term could include plant stomatal responses to $CO_2$ in theory. This is a very nice suggestion for us to try to characterize the plant stomatal responses to $CO_2$ using the catchment properties term in the future work, especially under high $CO_2$ condition.

For the question 3, we quite agree with your comments that there are other drivers of hydrological change in addition to climate change. Our method only considers the hydrological change due to climate change but neglects the effects of the variability of catchment properties (e.g., land cover change, groundwater and river water extraction, urbanization, irrigation, etc.) on the hydrology. According to your good comments, we made a discussion on the other drivers (catchment properties) of hydrological change for the interpretation of the results. For more information please see lines 363-368 in the revised MS.

**2.** The authors do acknowledge some of these issues to some extent at the end of the paper, but this is after the earlier discussion which often uses language of prediction, which I think goes too far. I would suggest terms such as "Climate change is projected to cause an increase (decrease) in R*∴∴.*" Also I suggest the authors address the above points in more detail, highlighting the limits to the interpretation of the CMIP5 ensemble in terms of likelihoods.

**Response:** Thank you very much for your nice comments. According to your good suggestions, we have changed the sentence "climate change will likely cause an obvious increase (decrease) of $R$…" to "climate change is projected to cause an increase (decrease) in $R$…". We also addressed the above points in more detail to highlight the limits to the interpretation of the CMIP5 ensemble in terms of likelihoods. For more information please see the red colored text in the revised MS.

**3.** My other concern is why the authors chose to use the Thorthwaite method for PET. It is stated on page 14 line 4 that this is because there is a "lack of meteorological data (such as relative humidity) in the GCM data. This is not true – GCMs are meteorological models, and indeed some of the CMIP5 GCMs are used in slightly different variants for numerical weather prediction. A huge range of meteorological outputs is available, including RH – see here http://cmip-pcmdi.llnl.gov/cmip5/docs/standard_output.pdf.

I recommend that the authors use the data portal http://cmippcmdi.llnl.gov/cmip5/data_description.html at PCMDI, who organised CMIP5. The Canadian Climate Centre webpage used by the authors only has a very limited number of variables.

**Response:** Thank you very much for your nice comments. In the original version (i.e. initial submitted manuscript), the *PET* of GCM for the baseline 1971–2000 and the future period 2071–2100 is estimated by the Thornthwaite method. We noted that the temperature-based Thornthwaite method is lack of physical basis, and it is necessary to justify the use of the Thornthwaite method and the use of more physically *PET* calculation methods. Thank you very much for informing us the meteorological data used for the *PET* calculation from the CMIP5 output http://cmippcmdi.llnl.gov/cmip5/data_description.html at PCMDI. Indeed, there is a huge range of meteorological outputs (including RH) from the CMIP5 models, which are enough for the calculation of *PET* by the Penman method. However, due to large amounts of data needed to be processed (including (1) download the 28 GCMs meteorological data, (2) statistical downscaling of the 28 GCMs meteorological data over China, (3) bias correction of the 28 GCMs meteorological data, (4) calculations of *PET* for the 28 GCMs, and (5) bias correction of *PET* for the 28 GCMs), so it is difficult for us to complete it in a short period. However, we tried our best to correct the *PET* of GCMs, and made a detailed comparison of the corrected *PET* method with other *PET* calculation methods to justify the use of *PET* calculation of the GCMs. In particular, there are three main changes for the *PET* calculations in the revised MS, which are as follows:

(1) We used a more physically *PET* data that estimated by the Penman equation (data during the period 1960–2008 provided by the Hydroclimatology Group of Princeton University) to calculate the climate elasticity over China instead of the *PET* data from the FAO Penman-Monteith method. We believe the climate elasticity would be more accurate in the revised MS than in the original version.

(2) We used a multiplicative correction method to correct the *PET* data of GCMs calculated from the Thornthwaite method (as shown in equation (1) in the revised MS). Based on the monthly data of temperature covering the period 1960–2008 provided by the Climatic Research Unit (CRU), the *PET* was calculated by the Thornthwaite method and then corrected by the equation (1) to test the applicability of the multiplicative correction method. The results indicated that the corrected annual *PET* shows a good agreement with that calculated by the Penman method (as shown in Figure 3 in revised MS). These two methods are quite consistent at both basin and grid scales, suggesting that the multiplicative correction method is acceptable for the bias correction of *PET* of the GCMs.

(3) We compared the four *PET* calculation methods (i.e., the Penman method, the Thornthwaite method, the FAO-56 Penman–Monteith method, and the Thornthwaite method corrected by the equation (1) in the revised MS) to test the robustness of the *PET* elasticity result subject to *PET* uncertainties. The results indicated that the mean annual *PET* by the Penman method, the FAO-56 Penman–Monteith method, and the Thornthwaite method corrected by the equation (1) are quite consistent, and the *PET* elasticity calculations from these three methods give very similar results in all 14 basins (as shown in Figure 13 in the revised MS). That is to say, the Thornthwaite method corrected by the equation (1) significantly improves the accuracy of *PET* and can be acceptable for the *PET* calculation of the GCMs.

Considering your good suggestions, in the future work we are going to calculate the Penman *PET* using the meteorological data from the CMIP5 output and further make a comparative analysis 
[revised manuscript text omitted]

---

## Author Response (AR2)

Dear Editor and Reviewers:

On behalf of all the contributing authors, I would like to express our sincere appreciations of your letter and reviewers' constructive comments concerning our article entitled "Responses of runoff to historical and future climate variability over China" (Manuscript No.: **hess-2017-98**). Those comments are all valuable and very helpful for revising and improving our paper, as well as the important guiding significance to our researches. We have studied comments carefully and have made correction which we hope meet with approval. In this revised version, changes to our manuscript were all highlighted within the document by using red colored text. Point-by-point responses to Referee #2 are listed below this letter.

**\*\*\*\*\*\*\*\*\*\*\*\*\*\*\*\*\*\*\*\*\*\*\*\*\*\*\*\*\*\*\*\*\*\*\*\*\*\*\*\*\*\*\*\*\*\*\*\*\*\*\*\*\*\*\*\*\*\*\*\*\*\*\*\*\*\*\*\*\*\*\*\***

**Anonymous Referee #2:**

1. I'm glad my previous comments were found to be useful. I can fully understand why the authors have not wished to repeat the future projections using the Penman method applied to the original CMIP5 projections (this would be a large undertaking) and to makes sense to try to use an approproximation for this by adjusting the Thornwaite-based results. It seems that this works reasonably well for the range of PET seen in the historical period. However, I think it is important to know how well this cover the range of PET projected for the future - if the maximum future PET is within or only slightly above the range tested in figure 3, then we can probably be confident that this method yields a good estimate of the what a Penman-based estimate would give for the future. However, if the future PET extends well above this range, the projections should therefore be viewed with more caution. Therefore I recommend that the authors give some information on the range of PET projected for the future, preferably shown on figure 3 in some way, so that the reader is aware of possible limitations of this approximation.

Response: Thank you for your nice comments. In the revised manuscript, we used a multiplicative correction method (equation (1)) to correct the *PET* that calculated from the Thornthwaite method. The results indicated that the corrected annual *PET* shows a good agreement with that calculated by the Penman method, suggesting that the multiplicative correction method is reasonable for predicting future *PET* of the GCMs (Figure 3). Figure 3 shows a comparison of annual *PET* of the period 1960–2008 calculated from the Penman method and the Thornthwaite method corrected by the multiplicative correction method. We failed to give some information of future annual *PET* of GCMs on Figure 3, since there are no estimates of future *PET* from the Penman method. However, we make a comparison of corrected mean annual *PET* from observation and the CMIP5 multi-model ensemble. As shown in Figure S1, the historical range of *PET* (1971–2000) from the CMIP5 multi-model ensemble shows a good agreement with the observational range (1960–2008). Under the future scenarios, the range of *PET* projected is slightly above that of the historical period due to climate warming. This further indicates that the multiplicative correction method used is reasonable for estimating future *PET* of the GCMs.

[Figure]

**Figure S1**. Box plots of mean annual *PET* for all 0.5° grid points over China during the observation period (1960–2008), historical simulation period 1971–2000 (His_sim) and future period (2071–2100) under three emission scenarios (RCP2.6, RCP4.5, and RCP8.5) of the CMIP5 multi-model ensemble. The *PET* is calculated from the Thornthwaite method and corrected by the multiplicative correction method.

2. I think it would also be worth explaining why Thornthwaite was used in the first place. e.g. is it following on form previous work?

Response: Thank you for your nice comments. We have made an explanation on the use of Thornthwaite method for predicting future *PET* in the revised manuscript (line 117).

3. In lines 345-346 of the revised manuscript, the authors say:

"Therefore, a more physically-basedbased PET calculation method (such as the Penman method) needs to be considered in the GCMs". I don't think "in the GCMs" is really what is meant here, as the authors are referring to their own calculations not those carried out *in* the GCMs - the GCM land surface schemes make their own calculations of ET, but those are not used here. The phrase "in the GCMs" should therefore be dropped from this sentence.

Response: Thank you for your nice comments. We have removed "in the GCMs" from this sentence.

4. Finally, a technical correction is needed, which I must admit I noticed first time but forgot to include in my previous review (apologies). In Table 1, the resolution of all the CMIP5 GCMs is given as 1*1 degree. However, this is not the resolution of the GCMs - I assume that it must be the resolution of the post-processed data obtained by the authors via the Canadian Climate Centre (presumably the data have been re-gridded from the native resolution to 1 degree resolution). This should be clarified.

Response: Thank you for your nice comments. The CMIP5 GCMs data, that we used, are statistically downscaled and regridded onto a common 1°×1° global grid by the Canadian Climate Data and Scenarios (CCDS). We have made a clarification for this point on the caption of Table 1 in the revised manuscript according to your good suggestions.

[revised manuscript text omitted]